# Arginine methylation of the DDX5 helicase RGG/RG motif by PRMT5 regulates resolution of RNA:DNA hybrids

Sofiane Y Mersaoui[1],[†] , Zhenbao Yu[1],[†] , Yan Coulombe[2,3], Martin Karam[1], Franciele F Busatto[2,3], Jean-Yves Masson[2,3,*] & Stéphane Richard[1,**]

## Abstract

Aberrant transcription-associated RNA:DNA hybrid (R-loop) formation often causes catastrophic conflicts during replication, resulting in DNA double-strand breaks and genomic instability. Preventing such conflicts requires hybrid dissolution by helicases and/or RNase H. Little is known about how such helicases are regulated. Herein, we identify DDX5, an RGG/RG motif-containing DEAD-box family RNA helicase, as crucial player in R-loop resolution. *In vitro*, recombinant DDX5 resolves R-loops in an ATP-dependent manner, leading to R-loop degradation by the XRN2 exoribonuclease. DDX5-deficient cells accumulate R-loops at loci with propensity to form such structures based on RNA:DNA immunoprecipitation (DRIP)-qPCR, causing spontaneous DNA double-strand breaks and hypersensitivity to replication stress. DDX5 associates with XRN2 and resolves R-loops at transcriptional termination regions downstream of poly(A) sites, to facilitate RNA polymerase II release associated with transcriptional termination. Protein arginine methyltransferase 5 (PRMT5) binds and methylates DDX5 at its RGG/RG motif. This motif is required for DDX5 interaction with XRN2 and repression of cellular R-loops, but not essential for DDX5 helicase enzymatic activity. PRMT5-deficient cells accumulate R-loops, resulting in increased formation of γH2AX foci. Our findings exemplify a mechanism by which an RNA helicase is modulated by arginine methylation to resolve R-loops, and its potential role in regulating transcription.

**Keywords** Arginine methylation; DDX5; RGG/RG motif; RNA helicase; XRN2
**Subject Categories** DNA Replication, Repair & Recombination; Post-translational Modifications, Proteolysis & Proteomics; RNA Biology
**The EMBO Journal (2019) 38: e100986**

## Introduction

In mammalian cells, there are nine protein arginine methyltransferases (PRMTs) responsible for transferring methyl groups from S-adenosyl methionine to the nitrogen atoms of arginine (Bedford & Clarke, 2009). Arginine methylation modulates multiple biological processes from gene expression, pre-mRNA splicing, the DNA damage response, and signal transduction (Blanc & Richard, 2017). PRMT5 is the major enzyme catalyzing symmetrical dimethylarginines, and as such, multiple responses have been observed upon inhibition of its function including the induction of the p53 response, DNA damage, and cell death. PRMT5 regulates gene expression by methylating histones and transcription factors (Friesen *et al*, 2001; Pal *et al*, 2004; Cho *et al*, 2012). Methylation of the Sm proteins by PRMT5 regulates the p53 response by modulating the alternative splicing of *MDM4* and *MDM2* (Bezzi *et al*, 2013). Inhibition of arginine methylation has been shown to induce DNA damage and sensitize cells to DNA-damaging agents, reviewed in Blanc and Richard (2017). PRMT5 regulates TIP60 activity by RUVBL1 methylation (Clarke *et al*, 2017) and by TIP60 alternative splicing to regulate the DDR (Hamard *et al*, 2018). This multitude of responses triggered in the absence of PRMT5 function suggests that it may be a good therapeutic target. Indeed, PRMT5 inhibition has been shown to decrease tumor growth in mouse models of mantle cell lymphoma, AML, CML, B-cell lymphoma, glioma, and breast cancer (Chan-Penebre *et al*, 2015; Koh *et al*, 2015; Li *et al*, 2015; Zhou *et al*, 2016; Braun *et al*, 2017; Hamard *et al*, 2018; Kaushik *et al*, 2018; Tan *et al*, 2019).

Physiologically, R-loops or RNA:DNA hybrids are programmed structures that occur during many cellular processes, including transcription, replication, and immunoglobulin class switching (Skourti-Stathaki & Proudfoot, 2014; Chedin, 2016; Bhatia *et al*, 2017). Persistent R-loops impede DNA replication and if unresolved ultimately cause DNA breaks and genomic instability (Hamperl & Cimprich, 2016; Aguilera & Gomez-Gonzalez, 2017) or

1   Departments of Oncology and Medicine, Segal Cancer Center, Lady Davis Institute for Medical Research, McGill University, Montréal, QC, Canada
2   Genome Stability Laboratory, Oncology Division, CHU de Québec-Université Laval, Québec, QC, Canada
3   Department of Molecular Biology, Medical Biochemistry and Pathology, Laval University Cancer Research Center, Québec, QC, Canada
    *Corresponding author. Tel: +1 418 525 4444 x 15154; E-mail: jean-yves.masson@crchudequebec.ulaval.ca
    **Corresponding author. Tel: +1 514 340 8260; E-mail: stephane.richard@mcgill.ca
    †These authors contributed equally to this work

mitochondrial instability (Silva *et al*, 2018). Therefore, it is not surprising that there are specialized machineries or protein complexes to prevent and resolve these R-loops. These structures can be unwound by RNA:DNA helicases, such as Senataxin (SETX) and Aquarius (AQR) (Skourti-Stathaki *et al*, 2011; Bhatia *et al*, 2014; Sollier *et al*, 2014; Hatchi *et al*, 2015), and the RNA in RNA: DNA hybrids degraded by RNase H1 and RNase H2 (Wahba *et al*, 2011). The RNA helicase Senataxin has been shown to resolve R-loops *in vitro*, and its deficiency in cells leads to R-loop accumulation (Skourti-Stathaki *et al*, 2011; Sollier *et al*, 2014; Hatchi *et al*, 2015). Senataxin is reported to function with the 5′–3′ exonuclease XRN2 to resolve a subset of R-loops at transcription termination sites of actively transcribed genes (Skourti-Stathaki *et al*, 2011; Morales *et al*, 2016; Aymard *et al*, 2017). The DNA helicase RECQ5 and RNA helicases DDX1 (Li *et al*, 2008, 2016; Ribeiro de Almeida *et al*, 2018), DDX19 (Hodroj *et al*, 2017), DDX21 (Song *et al*, 2017), DDX23 (Sridhara *et al*, 2017), and DHX9 (Cristini *et al*, 2018) were also found to be functionally involved in suppression of R-loops. Topoisomerase I removes the negative supercoils behind RNA polymerases to prevent annealing of the nascent RNA with the DNA template and suppresses R-loop formation (Tuduri *et al*, 2009). Fanconi anemia (FA) pathway proteins resolve RNA:DNA hybrids via FANCM translocase activity (García-Rubio *et al*, 2015; Schwab *et al*, 2015). Several RNA-processing proteins, such as the THO complex and the SRSF splicing factor, suppress R-loop formation, for example, by preventing the availability of the nascent RNAs for hybridization to template DNA (Huertas & Aguilera, 2003; Li & Manley, 2005; Paulsen *et al*, 2009; Wahba *et al*, 2011; Stirling *et al*, 2012; Sollier *et al*, 2014). The homologous recombination proteins, BRCA1 and BRCA2, are also involved in R-loop prevention and resolution (Bhatia *et al*, 2014; Hatchi *et al*, 2015). Predictably, mutations of proteins that prevent R-loop accumulation are frequently found in human diseases (Bhatia *et al*, 2017).

Despite these findings, little is known of the post-translational modifications regulating R-loop formation and resolution. It has been shown that the pausing of RNA polymerase II (RNA Pol II) increases DDX23 phosphorylation by SRPK2 enhancing R-loop suppression (Sridhara *et al*, 2017). Acetylation of DDX21 by histone acetyltransferase CBP regulates its helicase activity (Song *et al*, 2017), while methylation of RNA Pol II subunit POLR2A by PRMT5 regulates Senataxin recruitment at transcription termination regions (Zhao *et al*, 2016). Moreover, methylation of TDRD3 by CARM1 regulates the recruitment of topoisomerase IIIB at the c-Myc locus preventing negative supercoiling and R-loops (Yang *et al*, 2014). In this study, we define a new role for arginine methylation in the regulation of R-loops. We show that the methylation of the RNA helicase DDX5 by PRMT5 regulates its association with XRN2 to suppress R-loops at transcription termination regions and maintain genomic stability.

## Results

### DDX5 resolves R-loops *in vitro* and *in vivo*

DDX5 is an RGG/RG motif-containing helicase previously shown to unwind RNA:RNA and RNA:DNA duplexes (Hirling *et al*, 1989; Rossler *et al*, 2001; Xing *et al*, 2017). Dbp2, the *S. cerevisiae* homolog of DDX5, was shown to resolve RNA:DNA hybrids in the context of R-loops (Cloutier *et al*, 2016); however, whether DDX5 shares this R-loop resolving activity is unknown. To examine whether DDX5 resolves R-loops, we purified recombinant DDX5 to homogeneity (Fig 1A) and performed *in vitro* R-loop (RNA:DNA hybrid) and D-loop (DNA:DNA hybrid) unwinding activity assays using radiolabeled nucleotide substrates. The addition of increasing concentrations of DDX5 led to the appearance of a faster migrating species on native gels representing the DNA strands without the bound RNA fragment, and this occurred in an ATP-dependent manner (Fig 1B). Remarkably, DDX5 did not resolve the D-loop substrate (Fig 1B). These observations show that DDX5, like its yeast homolog Dbp2, resolves R-loops *in vitro*.

To examine whether DDX5 resolves R-loops *in vivo*, we generated DDX5-deficient U2OS cells using three siRNAs and we measured the accumulation of RNA:DNA hybrids by immunofluorescence using the monoclonal antibody (S9.6) known to recognize RNA:DNA hybrids within the mitochondria, nucleoli, and the nucleoplasm (Ginno *et al*, 2012; Bhatia *et al*, 2014; Sollier *et al*, 2014; Hamperl *et al*, 2017; Hodroj *et al*, 2017). We measured the S9.6 signal in the nucleoplasm as the total nuclear signal subtracting the nucleolar contribution. The nucleus and nucleolus were detected with DAPI and anti-nucleolin antibodies, respectively (Appendix Fig S1A). Depletion of DDX5 led to a significant increase of the S9.6 signal in the nucleoplasm of U2OS cells with all siRNAs tested (Fig 1C). We also visualized the accumulation of RNA:DNA hybrids from isolated genomic DNA using slot-blot analysis with the S9.6 antibody in the presence or absence of RNase H. Again, we observed a significant increase in the S9.6 signal within the genomic DNA isolated from DDX5-deficient cells compared to

---

**Figure 1.  DDX5 unwinds R-loops *in vitro* and represses cellular R-loop accumulation.**

A  Coomassie Blue staining of recombinant human DDX5 purified in bacteria. M denotes the molecular mass markers in kDa.

B  R-loop unwinding assay in the presence of increasing DDX5. The top panel shows a typical image obtained after the assay. The bar graph (bottom) shows the quantification. The average is expressed as percentage unwinding and standard error of the mean (SEM), n = 4.

C  S9.6 signal in the nucleoplasm of U2OS cells. An average from three independent experiments performed in triplicate. In total, ~90 images for each condition (10 images/slide and nine slides for each condition) were taken and two cells per image were quantified. Statistical significance was assessed using one-way ANOVA t-test. *P < 0.05; ****P < 0.0001.

D  U2OS cells transfected with siCTL or siDDX5 were subjected to DRIP-qPCR analysis with anti-IgG and anti-S9.6 antibodies with or without RNase H treatment. The gene location and genomic qPCR amplification region are shown at the top of each panel. B, E, H, S, and X denote the location of the *BsrG*I, *Eco*RI, *Hind*III, *Ssp*I, and *Xba*I. The identified R-loop peaks were extracted from the R-loop database (R-loop DB) for each region. The bar graphs are the average and SEM from three independent experiments. Statistical significance was assessed using t-test. *P < 0.05.

Source data are available online for this figure.

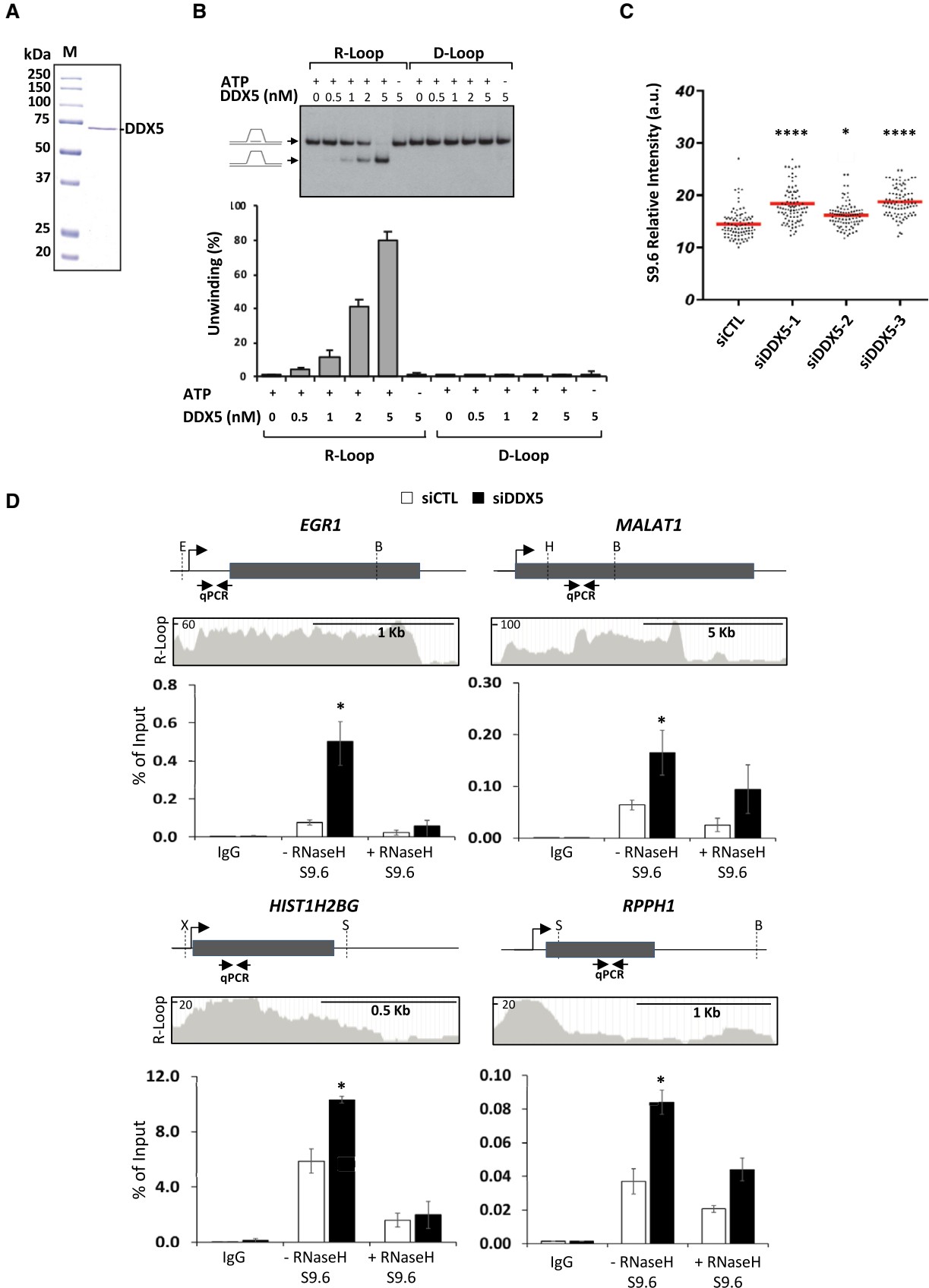

**Figure 1.**

control siRNA-treated cells. siSenataxin (siSTEX) was used as a positive control (Appendix Fig S1B and C).

To examine the role of the DDX5 in R-loop resolution at precise genomic loci, we used the RNA:DNA immunoprecipitation (DRIP) method. By qPCR analysis, we quantified R-loop accumulation at four selected loci, *EGR1, MALAT1, HIST1H2BG,* and *RPPH1*, previously known to have a propensity to form R-loops (Yang *et al,* 2014; García-Rubio *et al,* 2015; Hodroj *et al,* 2017). The R-loops for *EGR1* and *MALAT1* reside within the gene body, while the R-loops for *HIST1H2BG* and *RPPH1* encompass the transcription termination region, as determined previously and presented in the R-loop database (Fig 1D) (Wongsurawat *et al,* 2012). The knockdown of DDX5 resulted in a significant increase in R-loops at all four loci when compared to control siRNA conditions, and these were sensitive to RNase H treatment (Fig 1D). These findings suggest that DDX5-deficient cells accumulate R-loops.

### DDX5 deficiency leads to spontaneous DNA damage and hypersensitivity of U2OS cells to replication stress

As unresolved R-loops result in DNA damage (Skourti-Stathaki *et al,* 2011), we next investigated whether DDX5-deficient cells exhibit spontaneous DNA damage due to unresolved R-loops, and indeed, this was the case, as assessed by 1) an increase in the phosphorylation of H2AX termed γH2AX by immunoblotting (Appendix Fig S2A) and an increase in the number of γH2AX foci per nuclei (Appendix Fig S2B). Cells deficient in DDX5 are known to be sensitive to ionizing radiation (Nicol *et al,* 2013); however, it is unknown whether these cells are sensitive to DNA replication stress such as hydroxyurea, which lead to R-loop accumulation during DNA replication (Hamperl *et al,* 2017). Depletion of DDX5 using two different siRNAs led to a significant reduction of cell survival to hydroxyurea treatment compared with the control siRNA-transfected cells (Appendix Fig S2C). To further confirm this effect, we also performed a FACS (fluorescence-activated cell sorting)-based survival analysis of co-cultured cells (outlined in Appendix Fig S2D). This analysis enables direct comparison in the same culture of the proliferative fitness of DDX5-expressing and depleted cells. The U2OS cells without or with stable GFP expression were transfected with control and DDX5 siRNAs, respectively. The cells were mixed and co-plated 2 days after transfection and then treated with hydroxyurea. Compared to non-treated cells, a decrease in the percentage of GFP-positive cells after treatment indicates an increase in sensitivity to hydroxyurea in the target siDDX5-transfected cells (Appendix Fig S2E). Depletion of DDX5 caused significant increase in cell sensitivity to hydroxyurea, either at low dose (0.4 mM) with long-time incubation (24 h) or at high dose (10 mM) and short exposure (2 h; Appendix Fig S2F and G). Taken together, these results suggest that DDX5 deficiency causes cell hypersensitivity to the DNA replication stress agent hydroxyurea, a known R-loop inducer (Hamperl *et al,* 2017).

### DDX5 is arginine-methylated in U2OS cells

Interestingly, in a separate study, we identified DDX5 by mass spectrometry analysis as a PRMT5-interacting protein in two prostate cancer cell lines (Dataset EV1). DDX5 bears an arginine-rich motif, raising the possibility that DDX5 functions in R-loop homeostasis

are regulated by arginine methylation. Indeed, PRMT5 is known to catalyze the mono- and symmetrical dimethylation of arginine residues in proteins (Branscombe *et al,* 2001). We first confirmed the physical association between PRMT5 and DDX5 in U2OS cells. Cells were lysed, and either PRMT5 or DDX5 was immunoprecipitated followed by immunoblotting. Significant amounts of DDX5 were co-immunoprecipitated with anti-PRMT5 antibodies, but not control IgG (Fig 2A). The reverse was also observed, PRMT5 was co-immunoprecipitated with anti-DDX5 antibodies (Fig 2B). These findings confirm the association of PRMT5 with DDX5.

Next, to determine whether DDX5 is arginine-methylated, we first immunoprecipitated DDX5 from U2OS cells and analyzed it in the presence of methylarginines by mass spectrometry. We identified R502 within the C-terminal RGG/RG motif of DDX5 to harbor a mono-methyl (Fig 2C). U2OS cells transfected with Flag-epitope-tagged DDX5 were immunoprecipitated with anti-Flag antibodies, and the bound proteins immunoblotted with anti-methylarginine antibodies. We detected that Flag-DDX5 was arginine-methylated (Fig 2D).

### Deficiency of PRMT5 leads to increase in cellular R-loop accumulation

RGG/RG motifs are methylated mainly by PRMT1 and PRMT5, the two major enzymes responsible for cellular protein arginine methylation (Blanc & Richard, 2017). To identify which PRMT leads to R-loop accumulation, we turned to R-loop visualization using immunofluorescence assays with S9.6 antibody using siPRMT1 or siPRMT5 U2OS cells (Fig 2E). We observed that the knockdown of PRMT5 such as DDX5, but not PRMT1, led to significant increase of nuclear S9.6 staining in U2OS cells (Fig 2F). Furthermore, PRMT5-depleted cells had increased spontaneous DNA damage, as assessed by γH2AX intensity (Appendix Fig S3), consistent with the appearance of unsolved R-loops.

To further confirm that DDX5 and PRMT5 deficiencies cause R-loop accumulation, we used the DNA damage at RNA transcription (DART) system, which measures an RNA:DNA hybrid at a particular locus (Teng *et al,* 2018; Liang *et al,* 2019). The light-inducible chromophore-modified KillerRed (KR) is fused with either transcription activator (TA) or repressor (tetR). KR generates reactive oxygen species (ROS) through the excited chromophore and induces DNA damage and transcriptional activation at the genome-integrated tet response element (TRE) locus in the U2OS TRE cells. Elevated R-loop at the TRE locus over background is visualized by S9.6 immunofluorescence (Teng *et al,* 2018; Liang *et al,* 2019). DDX5 or PRMT5 knockdown (Fig 3A) led to a significant increase in R-loop specifically at the TA-KR marked locus, while the level of R-loops was similar to the control at the TetR-KR locus (Fig 3B and C). The signal we observe is similar to published studies (Teng *et al,* 2018). Under the conditions used (the staining involves a heating step on a 95°C heating block for 20 min to expose the antigen and blocking with 5% BSA), the S9.6 focus is more apparent than the typical nucleolar staining observed with standard S9.6 staining protocols. The fact that the cellular system is using a defined locus where multiple breaks are induced by KillerRed, this induces transcription very efficiently leading to R-loop accumulation and a stronger signal over typical nucleolar staining. These finding further confirm the accumulation of R-loops in the absence of DDX5 and PRMT5 using an independent assay, i.e., DART.

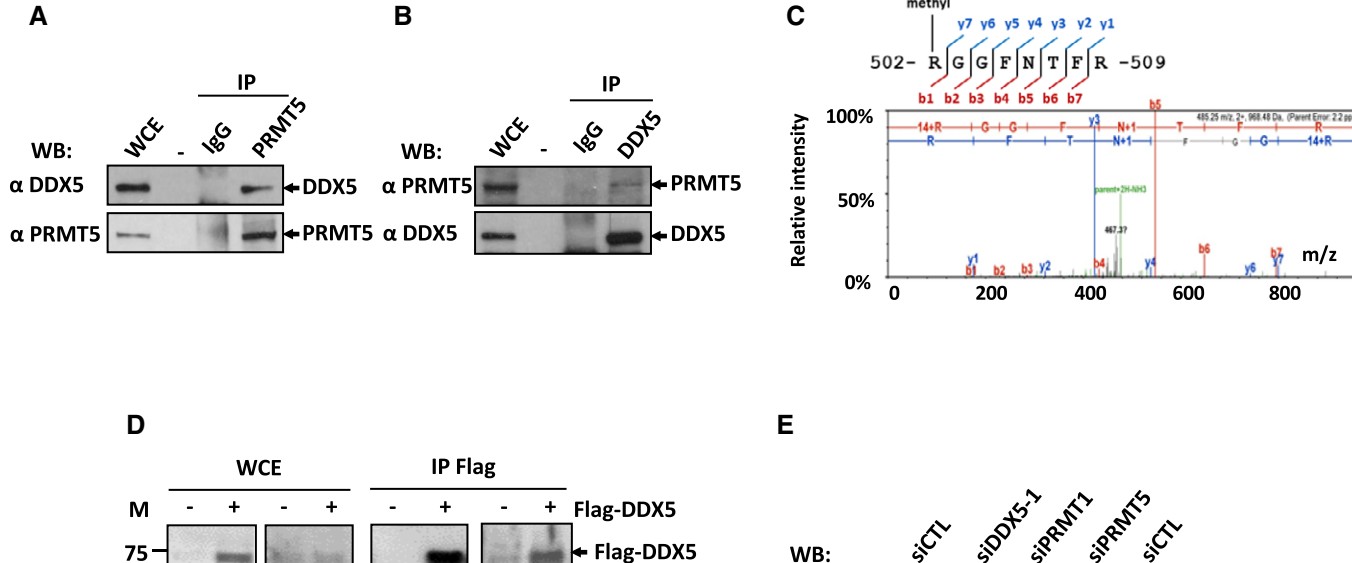

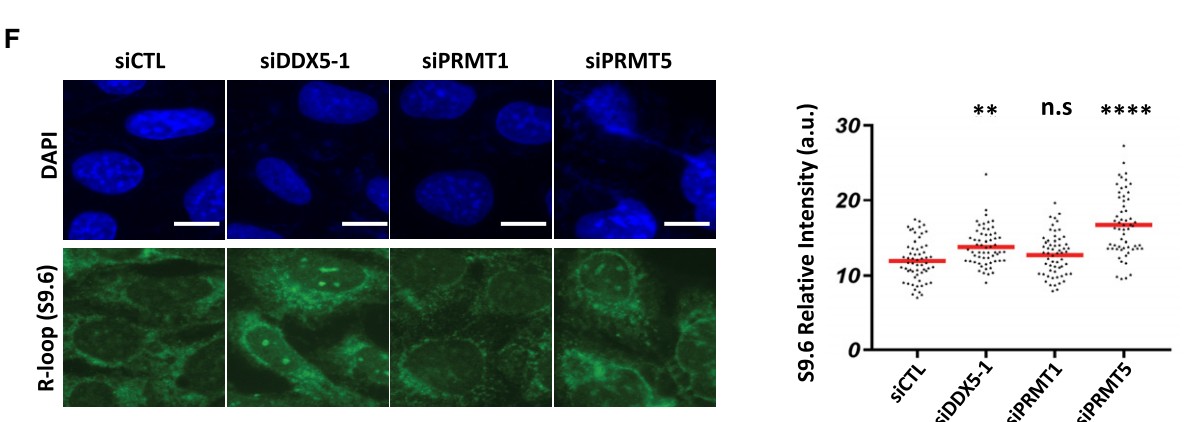

**Figure 2. DDX5 is arginine-methylated in U2OS cells, and PRMT5 deficiency causes cellular R-loops to accumulate.**

A  U2OS cells were lysed, and the lysates were incubated with anti-PRMT5 antibody or immunoglobulin G (IgG) as control followed by protein A-Sepharose bead addition. The immunoprecipitated (IP) proteins or whole cell extracts (WCEs) were separated by SDS–PAGE and Western-blotted (WB) with the indicated antibodies.

B  Same as panel (A) except anti-DDX5 antibody was used instead of anti-PRMT5 antibody for immunoprecipitation.

C  The mass spectrum of R502 monomethylation from Flag-DDX5 expressed in U2OS.

D  HEK293 cells transfected with Flag-DDX5 were lysed and immunoprecipitated with anti-Flag antibodies. The bound proteins were along with WCE from untransfected (−) and Flag-DDX5 (+)-transfected cells and were Western-blotted with anti-Flag and anti-methylarginine antibodies (MeR).

E, F  The knockdown using siRNAs of the indicated gene in U2OS cells was confirmed by Western blotting, and duplicate cell cultures were analyzed by immunofluorescence with S9.6 and anti-nucleolin antibodies. A typical image is shown. The scale bar represents 10 μm. Graphs show the average of 3 independent experiments performed in triplicates. Statistical significance was assessed using one-way ANOVA $t$-test. **$P < 0.01$ and ****$P < 0.0001$; n.s.: not significant.

Source data are available online for this figure.

We next performed DRIP-qPCR analysis with siPRMT5-transfected cells using the same four loci that showed accumulation of R-loops using siDDX5 (Fig 1D). PRMT5-depleted U2OS cells also displayed an increase in R-loops in an RNase H-dependent manner at the *EGR1, MALAT1, HIST1H2BG,* and *RPPH1* loci (Fig 4A). To further show that DDX5 and PRMT5 are linked in the same

**A**

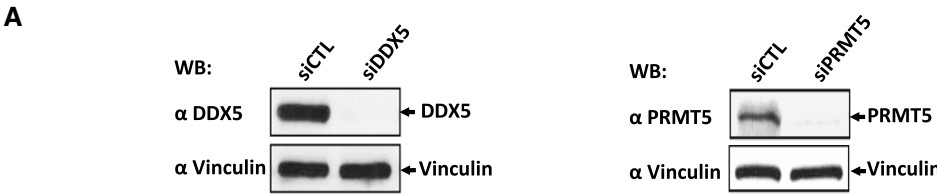

**B**

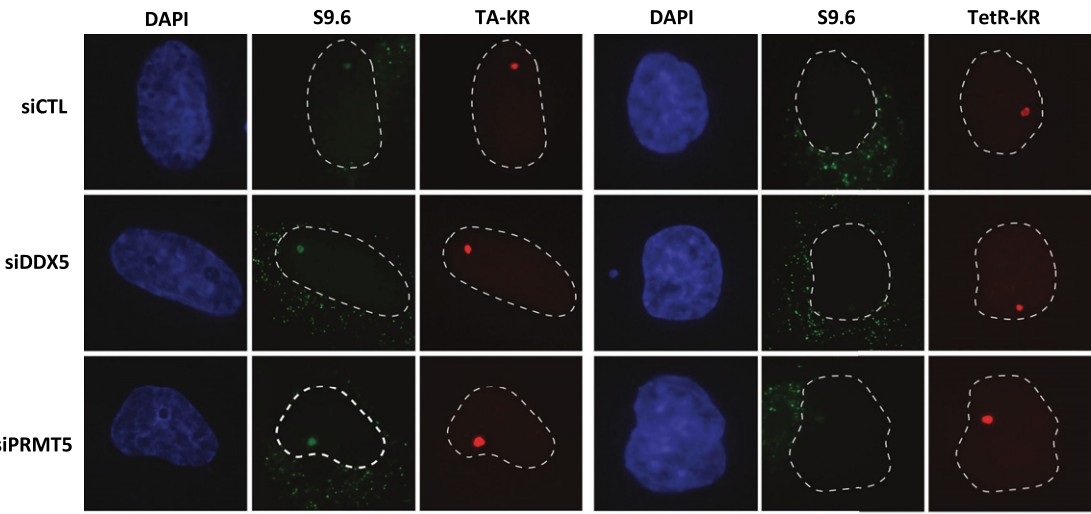

**C**

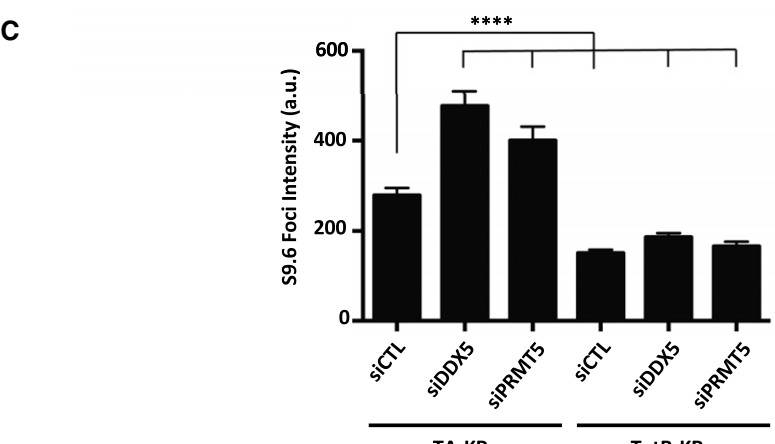

---

**Figure 3. Depletion of DDX5 increases R-loops at transcribed regions with local ROS-induced DNA damage.**

A  Western blotting of PRMT5 and DDX5 knockdowns.

B  Representative images of S9.6 staining in siCTL, siDDX5, or siPRMT5 knockdown at transcription on (TA-KR) or off (TetR-KR) genomic loci in U2OS TRE cells. The nuclei are visualized by DAPI.

C  Quantification of the S9.6 foci intensity in the indicated conditions. Three independent experiments were performed, and 50 cells were analyzed in each experiment. Error bars represent mean of S9.6 foci intensity quantification with SEM. The statistical analysis was performed by two-tailed Student's *t*-test (Mann–Whitney *U*-test). ****$P$ < 0.0001.

Source data are available online for this figure.

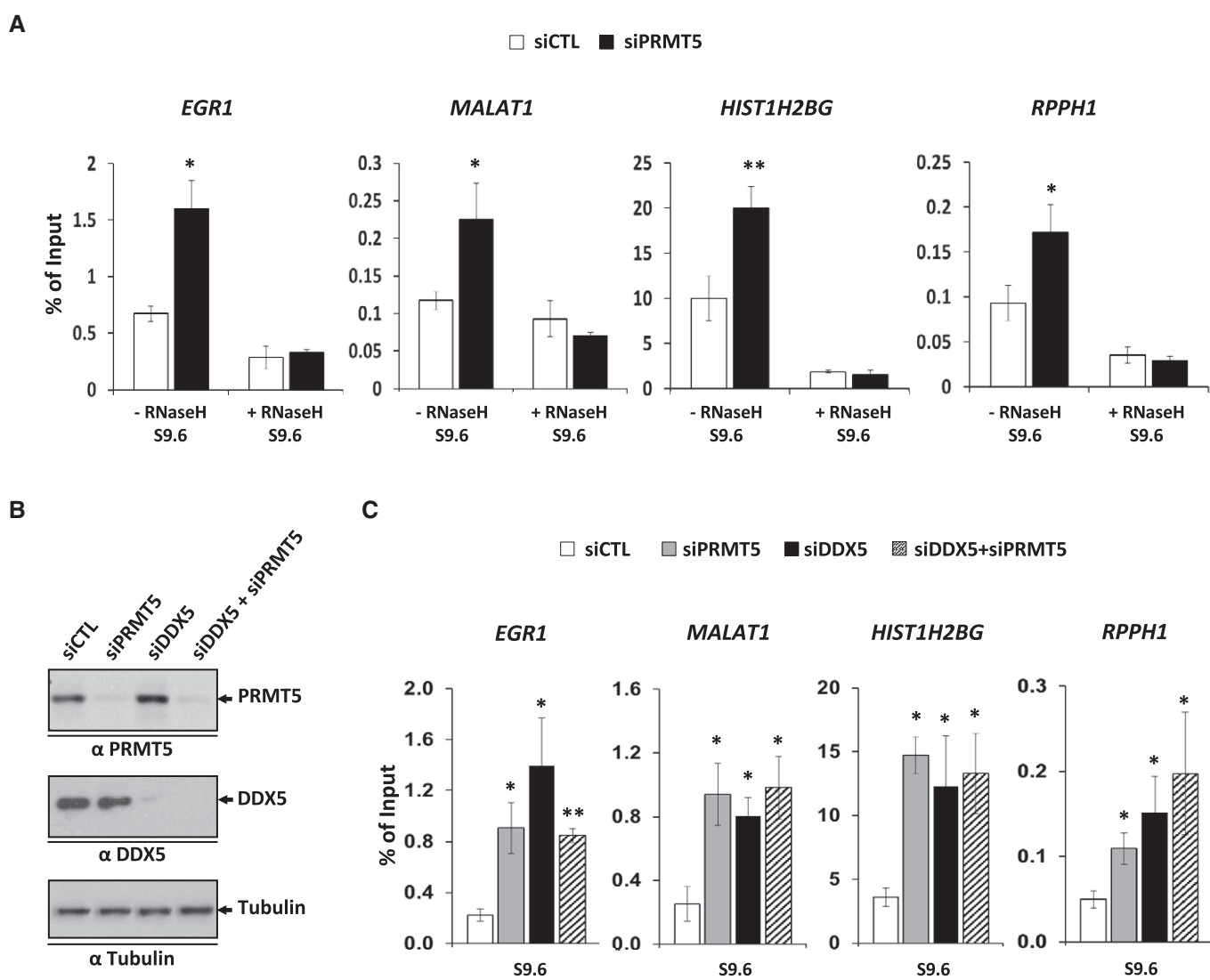

**Figure 4. PRMT5-deficient cells accumulate R-loops at specific loci.**

A  U2OS cells transfected with siCTL or siPRMT5 were subjected to DRIP-qPCR analysis. The average and SEM from three independent experiments are shown. Statistical significance was assessed using Student's *t*-test. *$P < 0.05$ and **$P < 0.01$.

B  Western blotting of whole cell extracts from siCTL-, siPRMT5-, siDDX5-, or siDDX5/ siPRMT5-transfected cells.

C  U2OS cells transfected with siCTL, siPRMT5, siDDX5 or siDDX5/ siPRMT5 were subjected to DRIP-qPCR analysis. The average and SEM from three independent experiments are shown. Statistical significance was assessed using Student's *t*-test. *$P < 0.05$ and **$P < 0.01$.

Source data are available online for this figure.

pathway, we performed a double knockdown (Fig 4B) and assessed R-loops at the same loci. The double depletion did not reveal a synergistic increase in R-loop accumulation than the single depletion of either PRMT5 or DDX5 (Fig 4C), suggesting PRMT5 and DDX5 are functionally linked for this function.

**DDX5 is methylated by PRMT5 at its C-terminal RGG motif**

Flag-DDX5 was expressed in U2OS cells, and anti-Flag immunoprecipitations were separated by SDS–PAGE and the bound proteins immunoblotted with anti-symmetrical dimethylarginine antibodies (SMDAs). We observed that PRMT5 knockdown caused a significant

reduction in the symmetrical arginine dimethylation of DDX5 (Fig 5A). These findings confirm that DDX5 is indeed an *in vivo* substrate of PRMT5. To map the methylated region, we next performed *in vitro* arginine methylation analysis using glutathione-S-transferase (GST)–DDX5 fusion proteins. DDX5 has two RGG/RG motifs: One located at its N-terminus and another at its C-terminus. Interestingly, both DDX5 RGG/RG motifs are conserved in the yeast homolog Dbp2 (Fig 5B). We generated three truncation mutants of DDX5, including the N-terminal region (residues 1–100; F1), the central catalytic enzyme domain (92–471; F2), and the C-terminal region (residues 466–614; F3; Fig 5C). Only the C-terminal region (F3), encompassing the RGG/RG motif, was methylated by PRMT5

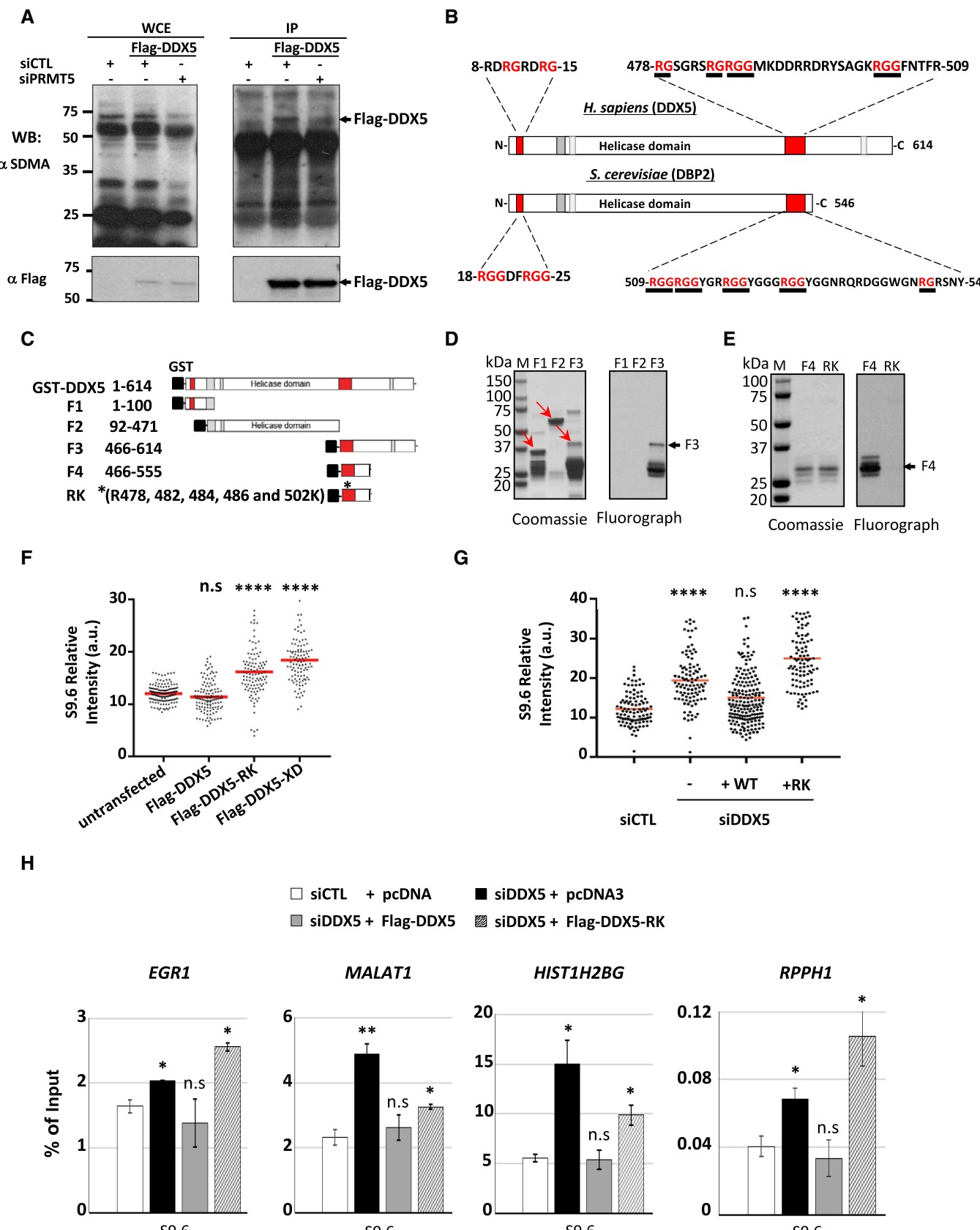

Figure 5.

◀

**Figure 5. The RGG/RG motif of DDX5 is a substrate for PRMT5 and is required for R-loop resolution *in vivo*.**

A    U2OS cells were transfected with a Flag-DDX5 expression vector in the presence of siCTL or siPRMT5. Lysates were immunoprecipitated with IgG or anti-Flag antibodies as indicated. The WCE and the bound proteins were Western-blotted with anti-symmetrical dimethylarginine antibody or anti-Flag antibodies.

B    A schematic of DDX5 helicase domain and RGG/RG motifs is shown.

C    GST fusion of DDX5 fragments (F1-F4) and the RK mutant used in this study. RK (*) indicates R to K substitutions of 5 arginines (R478, R482, R484, R486, and R502).

D, E  Coomassie Blue (left panel) and *in vitro* methylation assay (right panel) of indicated GST-DDX5 fragments and the GST-DDX5-RK mutant.

F    Immunofluorescence analysis with S9.6 and anti-Flag antibodies of U2OS cells transfected with Flag-tagged DDX5, DDX5-RK, or DDX5-XD (helicase dead). Nuclear S9.6 signal was counted in both Flag-negative and Flag-positive cells. The Flag-negative cells were considered as untransfected cells. The graphs shown represent the quantification with the SEM from three independent experiments. Statistical significance was assessed using one-way ANOVA *t*-test. ****$P < 0.0001$.

G    Immunofluorescence analysis with S9.6 and anti-Flag antibodies of U2OS cells transfected with siCTL or siDDX5-1 and Flag-tagged DDX5 (+WT) or DDX5-RK (+RK) as indicated. The graphs show the average and SEM from three independent experiments. Statistical significance was assessed using one-way ANOVA *t*-test. ****$P < 0.0001$.

H    HEK293 cells were transfected with siCTL or siDDX5. The next day, the cells that received siDDX5 were subsequently transfected with empty pcDNA3 vector (−), expression vectors encoding Flag-DDX5 (WT), or Flag-DDX5-RK (RK). Forty-eight hours later, all the cells were subjected to DRIP-qPCR analysis. The average with the SEM from three independent experiments is shown. Statistical significance was assessed using Student's *t*-test. *$P < 0.05$ and **$P < 0.01$; n.s.: not significant.

Source data are available online for this figure.

(Fig 5D). We then substituted DDX5 R478, R482, R484, R486, and R502 within the RGG/RG motif with lysines in a smaller region (466–555; F4). The 5R to 5K mutation within the F4 fragment of DDX5 (RK) completely abolished arginine methylation by PRMT5 (Fig 5E).

### DDX5 helicase activity and its C-terminal RGG/RG motif are required for cellular R-loop suppression

Next, we examined whether R to K substitutions of DDX5 affected its R-loop resolution function. Interestingly, ectopic expression of DDX5-RK as well as the catalytically inactive mutant, DDX5-XD, caused significant increase of R-loop accumulation in U2OS cells, as determined by S9.6 immunofluorescence in the Flag-positive cells (Fig 5F and Appendix Fig S4A). These findings suggest that the ectopic expression of Flag-DDX5-RK and Flag-DDX5-XD likely behave as dominant negatives, unlike Flag-DDX5 (Fig 5F). We further investigated whether WT DDX5 or DDX5-RK could rescue the R-loop accumulation observed in siDDX5 cells. The ectopic expression of WT DDX5 rescued significantly the R-loop accumulation observed in siDDX5 cells; however, this was not observed with the expression of DDX5-RK (Fig 5G and Appendix Fig S4B). We also performed a rescue experiment and measured R-loops at specific loci using DRIP-qPCR analysis. As a high transfection efficiency of plasmid DNA is required for rescue analysis by DRIP-qPCR, we used HEK293 cells. Similarly, as was observed in U2OS cells, depletion of DDX5 also caused a significant increase of R-loop accumulation in HEK293 cells. As expected, the Flag-DDX5 restored the effect of siDDX5 siRNA on R-loop accumulation at all loci analyzed (Fig 5H). By contrast, the Flag-DDX5-RK mutant fully reversed the effects of siDDX5 at *EGR1* and *RPPH1* loci, and partially reversed the siDDX5 effects at *MALAT1* and *HIST1H2BG* loci (Fig 5H). These results suggest that the RGG/RG motif is required for the regulation of DDX5 function in cellular R-loop resolution.

### DDX5 associates with known R-loop regulatory proteins

To define the mechanism by which arginine methylation regulates DDX5 function in cellular R-loop suppression, we performed *in vitro* R-loop unwinding assays using purified WT DDX5 and DDX5-RK from insect cells. As shown in Appendix Fig S5, substitution of arginine with lysine at the RGG motif did not affect DDX5 helicase

activity, suggesting that instead of modulating its enzymatic activity, the RGG motif may have other roles for the regulation of DDX5 function in the cellular R-loop suppression, as, for example, in mediating protein–protein interaction. We then performed stable isotope labeling with amino acids in cell culture (SILAC)-based mass spectrometry analysis to identify interacting partners, which may regulate DDX5 function in R-loop metabolism. U2OS cells expressing Flag-DDX5 were grown in the "heavy" medium and the control (pcDNA3) U2OS cells grown in "light" medium (Appendix Fig S6A). Many RNA binding proteins belonging to the heterogeneous nuclear ribonucleoproteins (hnRNPs) and DEAD/DEAH-box families were identified (Dataset EV2 and Appendix Fig S6B). These included known DDX5 interactors including DHX9, DDX3X, and DDX17 (Ogilvie *et al*, 2003; Wilson & Giguere, 2007; Choi & Lee, 2012). Interestingly, we also identified proteins known to influence R-loop resolution: DDX1, DHX9, XRN2, SRSF1, and Aly/REF, an exon junction complex protein (Dataset EV2). Of these, XRN2 caught our attention as it functions with helicases such as Senataxin (Skourti-Stathaki *et al*, 2011; Cristini *et al*, 2018) and DHX9 (Cristini *et al*, 2018) and is required to resolve a subset of R-loops (Morales *et al*, 2016).

### RGG/RG motif of DDX5 mediates association with XRN2

Hence, we hypothesized that XRN2 also assists DDX5 with R-loop resolution of a subset of genes. We first confirmed the interaction between the two proteins. U2OS cells expressing Flag-DDX5 were immunoprecipitated with anti-Flag antibodies, and the presence of XRN2 was detected by immunoblotting (Fig 6A), confirming our mass spectrometry data. Endogenous DDX5 also co-immunoprecipitated with XRN2, but not control IgG (Fig 6B). We next mapped the region of DDX5 required to associate with XRN2. DDX5 is composed of a central helicase domain with N- and C-terminal RGG/RG motifs (Fig 5B). The XRN2 interaction region was mapped between amino acids 435 and 554 of DDX5, where the C-terminal RGG/RG motif resides (Fig 6C). We next determined whether mutation of the arginines within the C-terminal RGG/RG motif affected recognition with anti-methylarginine antibody and interaction with XRN2. Indeed, Flag-DDX5-RK was not recognized with the anti-methylarginine antibody and had reduced association with XRN2, unlike WT Flag-DDX5 (Fig 6D). These findings suggest that arginine methylation of the RGG/RG motif of DDX5 regulates interaction with XRN2.

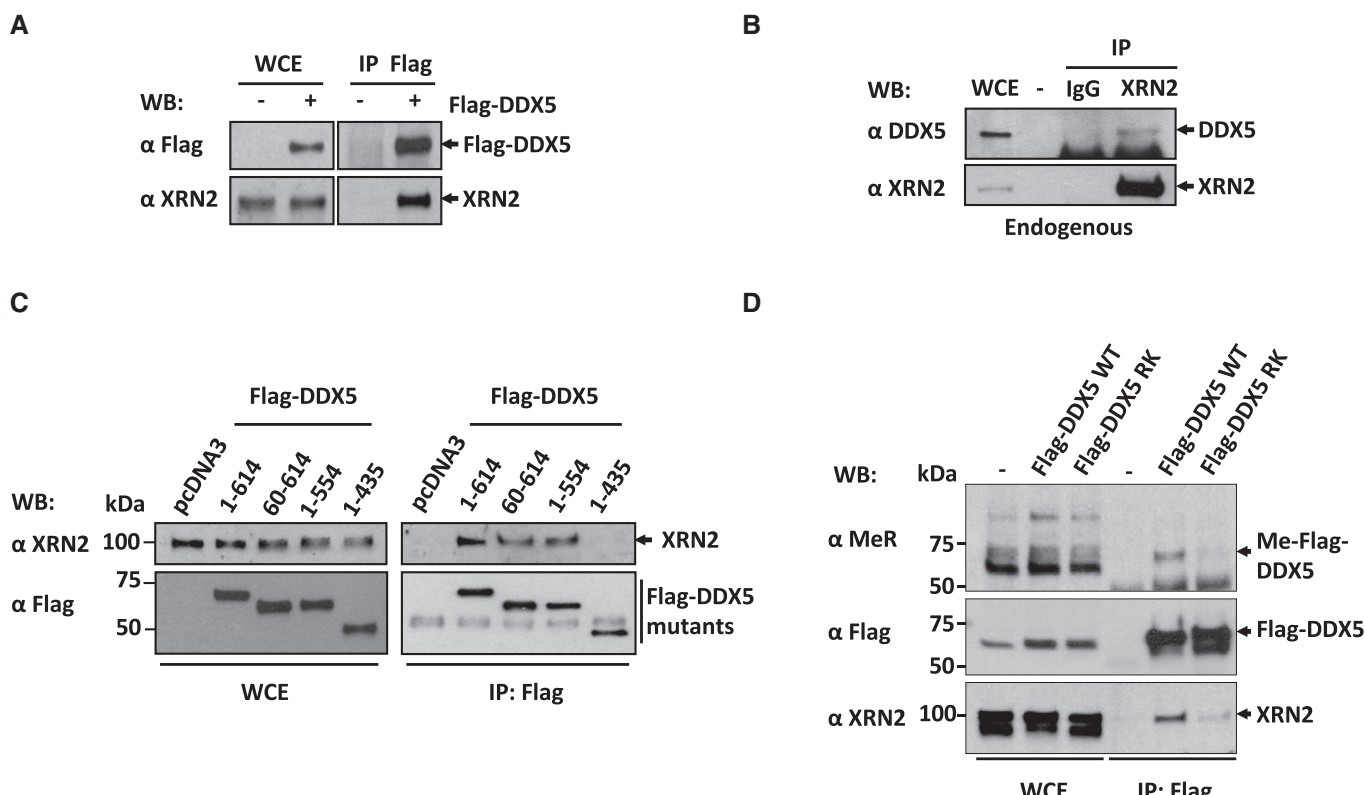

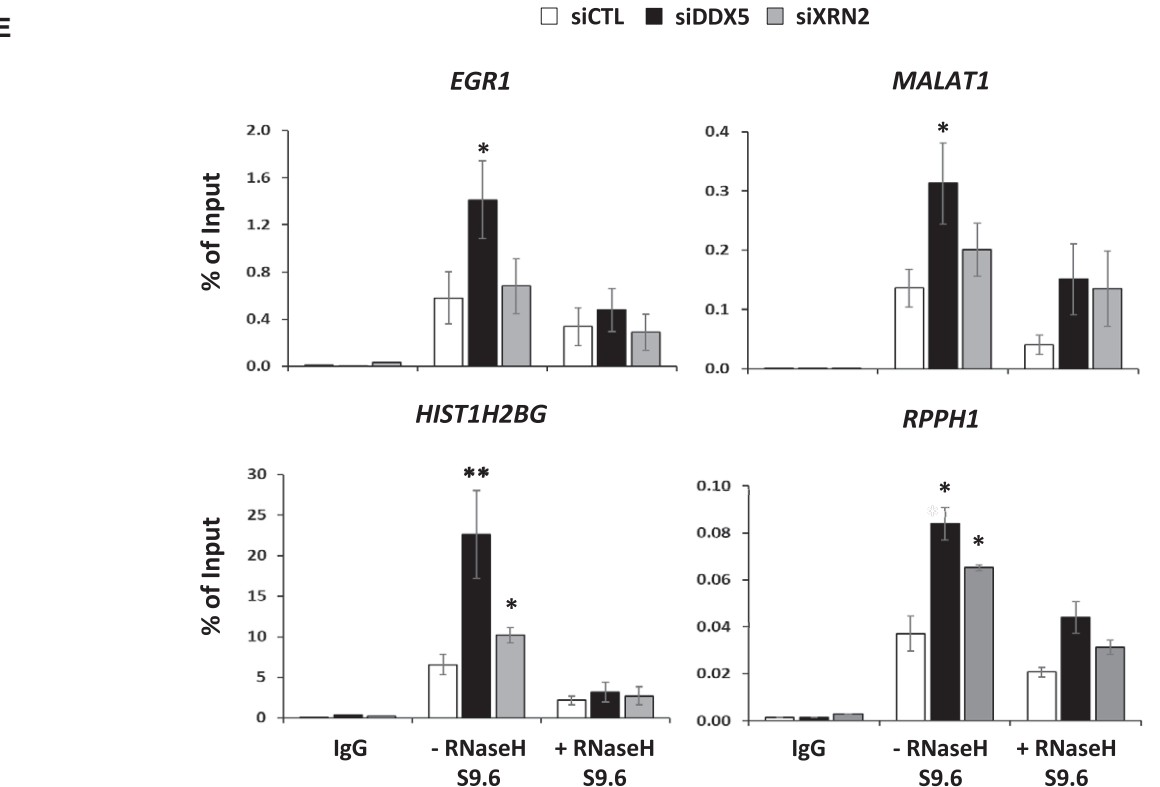

Figure 6.

**Figure 6.  DDX5 physically associates with the 5′–3′ exonuclease XRN2, functioning together to repress R-loops.**

A   HEK293 cells were transfected with empty pcDNA3 vector (−) or Flag-DDX5 (+). Whole cell extracts (WCEs) and anti-Flag immunoprecipitations (IPs) were immunoblotted with anti-Flag and anti-XRN2 antibodies.

B   Untransfected U2OS cells were lysed and subjected to immunoprecipitation with control immunoglobulin G (IgG) or anti-XRN2 antibodies. The bound proteins were separated by SDS–PAGE followed by Western blotting with anti-DDX5 or anti-XRN2 antibodies. The migration of DDX5 and XRN2 is depicted with an arrow.

C   U2OS cells were transfected with empty pcDNA3 vector (pcDNA3) or expression vectors encoding Flag-DDX5 WT (1–614) or truncated Flag-DDX5 proteins (60–614; 1–554; 1–435). WCEs of the transfected cells (left panel) were Western-blotted with anti-Flag and anti-XRN2 antibodies to confirm equivalent expression. The transfected cells were lysed, and anti-Flag immunoprecipitations (right panel) performed in the presence of co-immunoprecipitating endogenous XRN2 by Western blotting with anti-XRN2 antibodies. M denotes molecular mass markers in kDa.

D   Untransfected and U2OS cells stably expressing Flag-DDX5 WT or Flag-DDX5-RK were subjected to immunoprecipitation with anti-Flag antibody. The WCEs and the anti-Flag immunoprecipitated proteins were Western-blotted with anti-monomethylarginine (MeR), anti-Flag, and anti-XRN2 antibodies, respectively. M denotes molecular mass markers in kDa.

E   U2OS cells transfected with siCTL, siDDX5, or siXRN2 were subjected to DRIP-qPCR analysis with anti-IgG and anti-S9.6 antibodies with or without RNase H treatment. The average and SEM from three independent experiments are shown. Statistical significance was assessed using Student's *t*-test. *$P < 0.05$; **$P < 0.01$.

Source data are available online for this figure.

## DDX5 and XRN2 function together in R-loop resolution

To test whether DDX5 and XRN2 function together in R-loop resolution, we performed DDX5 R-loop resolving *in vitro* assays in the presence or absence of purified XRN2 (Appendix Fig S7A). The presence of recombinant XRN2 (20 nM) increased the RNase product (i.e., degraded RNA) component of the RNA:DNA hybrid unwound by DDX5 (Appendix Fig S7B). However, XRN2 did not influence the overall ability of DDX5 to resolve R-loops (Appendix Fig S7B). XRN2 requires a 5′-phosphate end to allow the exonuclease activity to proceed in the 5–3′ direction (Stevens & Poole, 1995). Thus, in our R-loop substrate, the 5′-P-end was not protuberant and might not have been accessible for RNA degradation. To exclude this possibility, we designed new substrates adding 13-base non-complementary 5′ overhang protruding from the R-loop. An excess of purified XRN2 (100–400 nM) on the 5′ overhang R-loop substrate (100 nM) leads to a progressive degradation of the RNA, leading to faster migrating forms of the substrate (Appendix Fig S7C, left). At 400 nM XRN2, the resulting product had a slower mobility than a R-loop with complementary RNA (our original substrate, lane 5). This infers that XRN2 could not degrade completely the RNA and stopped at the RNA:DNA junction. To recapitulate this result, we used a simpler RNA:DNA duplex with the same protuberating 5′-P-end (Appendix Fig S7C, right). Similarly, XRN2 could not degrade completely the RNA, leading to a higher migrating form than the annealed RNA:DNA duplex without the 5′ overhang (lane 5). Altogether, these results support our previous experiments (Appendix Fig S7B) where DDX5 needs to unwind the R-loop to allow complete degradation of the RNA by XRN2.

Immunofluorescence and slot-blot analysis using S9.6 revealed XRN2 deficiency led to an R-loop accumulation in U2OS cells (Appendix Fig S8A–C), as previously reported (Morales *et al*, 2016). To study the functional relationship between DDX5 and XRN2, we analyzed the effect of XRN2 deficiency on R-loop accumulation at the same four loci as performed above in DDX5-deficient cells. XRN2 knockdown caused an R-loop accumulation at two of the four loci (Fig 6E; *HIST1H2BG* and *RPPH1*). Interestingly, the R-loops we amplified, between indicated restriction sites (Fig 1D), corresponded to the areas of RNA:DNA hybrids extending over the termination pause sites, while R-loops in the *EGR1* and *MALAT1* loci were located within their gene bodies and were not significantly enriched in the absence of XRN2 (Fig 6E). These results suggest that DDX5 functions with XRN2 to resolve R-loops at transcription termination

sites, but also has additional XRN2-independent functions in R-loop suppression within gene bodies. Transcription is required for the creation of R-loops and thus to exclude that deficiency of DDX5, XRN2, and PRMT5 upregulates gene expression and thus R-loop as a secondary event, we measured the mRNA levels of *EGR1*, *MALAT1*, *HIST1H2BG*, and *RPPH1*. The knockdown of DDX5, XRN2, and PRMT5 in U2OS cells did not significantly affect gene expression of the four loci, except for an increase of *MALAT1* expression in PRMT5-deficient cells (Appendix Fig S9). Thus, this rules out an increase in transcription as the cause of R-loop accumulation in DDX5-, XRN2-, and PRMT5-depleted cells.

### DDX5 function in R-loop resolution is associated with the release of RNA Pol II from transcriptional termination regions

The activity of the XRN2 exonuclease is required to promote transcriptional termination of a subset of genes (Kim *et al*, 2004; West *et al*, 2004). Formation of R-loops at transcriptional termination regions is a requirement for RNA Pol II pausing downstream of the poly (A) site at certain loci, and the activity of XRN2 is required to facilitate termination (Skourti-Stathaki *et al*, 2011; Cristini *et al*, 2018). To examine the role of DDX5 and to compare it to XRN2, we chose six RNA Pol II-transcribed genomic loci and performed DRIP-qPCR and RNA Pol II ChIP at their transcriptional termination regions in the absence of DDX5 or XRN2. Three loci (*PRMT7*, *LINC01346*, and *NFKBIL2*) were selected due to their known R-loop formation at the transcriptional terminal regions from the R-loop database (Wongsurawat *et al*, 2012), and three loci were selected because they are known to be regulated by XRN2 (*SLC25A3*, *JUN*, and *EEF1A1*) (Fong *et al*, 2015). U2OS cells transfected with siDDX5 or siXRN2 were prepared for DRIP-qPCR and RNA Pol II by ChIP-qPCR analysis. In all cases examined (*PRMT7, LINC01346, NFKBIL2, SLC25A3, JUN,* and *EEF1A1*), increased R-loops were detected by S9.6 DRIP-qPCR in either XRN2- or DDX5-depleted cells (Fig 7). Furthermore, RNA Pol II accumulated at transcription termination pause sites of these 6 loci in XRN2- or DDX5-deficient cells (Fig 7). These findings demonstrate that DDX5, such as XRN2, is required for R-loop resolution at transcription termination sites to facilitate RNA Pol II release at certain loci.

RNA Pol II distribution, R-loop propensity, and XRN2-mediated transcription termination have been well demonstrated along the *β-actin* gene (Fig 8A) (Kaneko *et al*, 2007; Skourti-Stathaki *et al*, 2011; Zhao *et al*, 2016; Cristini *et al*, 2018). HEK293 cells

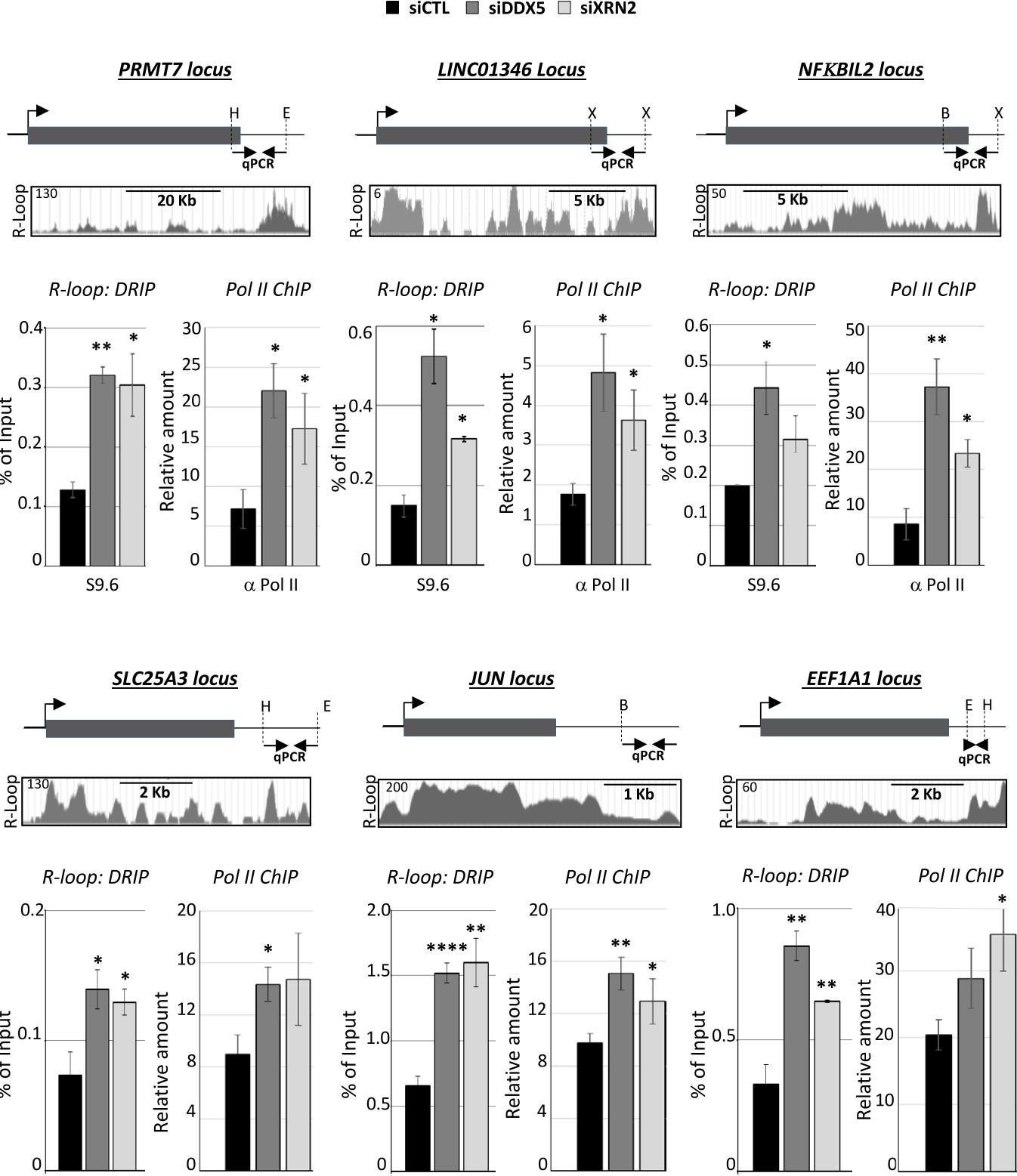

**Figure 7.  Depletion of DDX5 or XRN2 causes enrichment of R-loops and RNA Pol II at the transcriptional termination pause site.**
HEK293 cells were transfected with siCTL, siDDX5, or siXRN2. The gene location and qPCR amplification region are shown at the top of each panel. The identified R-loop peaks were extracted from R-loop database (R-loop DB) for each gene region. *R-loop: DRIP* denotes anti-S9.6 DRIP-qPCR, and *Pol II ChIP* denotes RNA Pol II ChIP. The bar graphs show the average and SEM from at least three independent experiments. Statistical significance was assessed using Student's *t*-test. *P < 0.05; **P < 0.01; and ****P < 0.0001. B, E, H, S, and X denote the location of the *BsrG*I, *Eco*RI, *Hind*III, *Ssp*I, and *Xba*I.

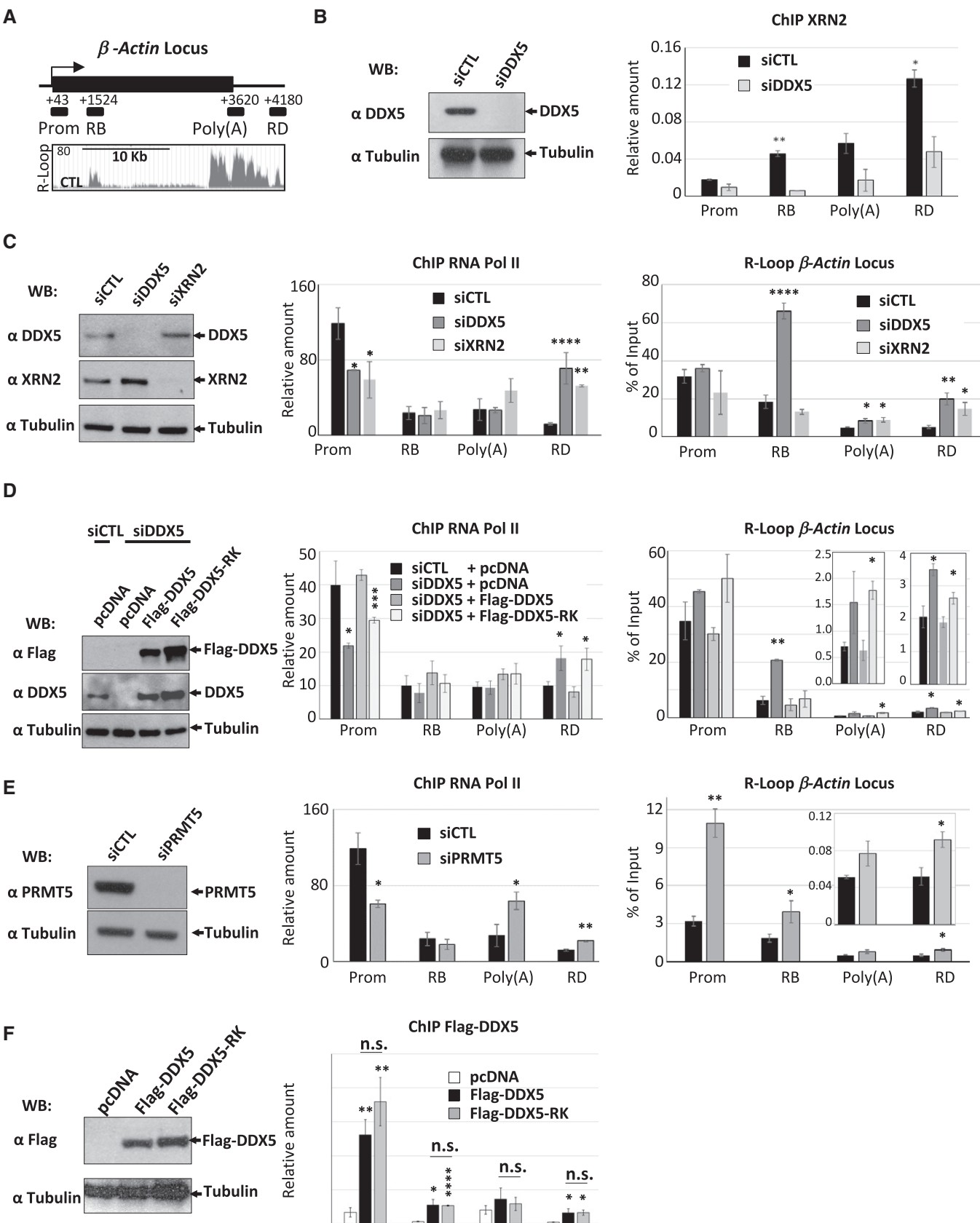

Figure 8.

◀

**Figure 8.  DDX5 cooperates for XRN2 recruitment and facilitates the release of RNA Pol II from β-actin transcriptional termination pause site.**

A  Schematic representation of the β-actin locus and the location of the genomic PCR primers used for ChIP-qPCR and DRIP-qPCR. Prom (promoter), RB (region B), and poly(A) and RD (region D) indicate the regions analyzed by qPCR.

B  HEK293 cells were transfected with siCTL or siDDX5, and XRN2 ChIP assays were performed. The XRN2 ChIP signal was normalized to the Pol II ChIP signal. WCEs were immunoblotted with anti-DDX5 and α-tubulin antibodies, the latter was used as a loading control. The graphs show the average and SEM from at least three independent experiments. Statistical significance was assessed using Student's t-test. *P < 0.05; **P < 0.01.

C  HEK293 cells were transfected with siCTL, siDDX5, or siXRN2. The cells were subjected to Western blotting, RNA Pol II ChIP, and DRIP analyses of β-actin gene. The graphs show the average and SEM from at least three independent experiments. Statistical significance was assessed using Student's t-test. *P < 0.05; **P < 0.01; and ****P < 0.0001.

D  HEK293 cells were transfected with siCTL or siDDX5. The next day, the cells that received siDDX5 were subsequently transfected with empty pcDNA3 vector (−), expression vectors encoding Flag-DDX5 (WT), or Flag-DDX5-RK (RK). The cells were subjected to Western blotting to visualize Flag-DDX WT and RK expression. Pol II ChIP and DRIP analyses of β-actin gene were performed. For the ChIP analysis, the Y-axis shows the signal-to-noise ratio of RNA Pol II IP relative to control IgG IP. The graphs show the average and SEM from at least three independent experiments. Statistical significance was assessed using Student's t-test. *P < 0.05; **P < 0.01; and ****P < 0.0001.

E  HEK293 cells were transfected with siCTL or siPRMT5. The cells were subjected to Western blotting, RNA Pol II ChIP, and DRIP analyses of β-actin gene. The graphs show the average and SEM from at least three independent experiments. Statistical significance was assessed using Student's t-test. *P < 0.05 and **P < 0.01.

F  HEK293 cells were transfected with empty plasmid (−), wild-type Flag-DDX5 (WT), or its RK mutant (RK) and subjected to Western blot and Flag-DDX5 ChIP-qPCR analysis of β-actin gene. The graphs show the average and SEM from at least three independent experiments. Statistical significance was assessed using Student's t-test. *P < 0.05; **P < 0.01; and ****P < 0.0001; n.s. not significant.

Source data are available online for this figure.

transfected with siCTL showed XRN2 distribution by ChIP analysis throughout the β-actin locus with higher levels at the region D (RD) 3′ of the poly (A) site, as previously reported (Kaneko et al, 2007; Skourti-Stathaki et al, 2011). A significant decrease in XRN2 occupancy was observed in siDDX5-depleted cells (Fig 8B; region B, poly (A) and RD, right panel). These findings suggest that DDX5 regulates the recruitment of XRN2 at β-actin locus. Furthermore, DDX5 depletion caused a dramatic increase in RNA Pol II deposition at the β-actin terminator region (RD) and a decrease in the promoter region (Prom), as observed in XRN2-depleted cells (Fig 8C, middle panel), indicating a redistribution of RNA Pol II away from promoter-proximal position to transcriptional termination region. DRIP-qPCR analysis confirmed a significant increase in R-loop accumulation at the poly(A) site and the downstream termination region in DDX5- and XRN2-deficient cells (Fig 8C, right panel, poly (A) and RD). Interestingly, knockdown of DDX5, but not XRN2, also caused significant increase in R-loop at β-actin body (RB), further confirming that DDX5 also resolves R-loops at XRN2-independent regions, i.e., gene body.

We then transfected siRNA-resistant Flag-DDX5 and Flag-DDX5-RK into siDDX5-depleted HEK293 cells and performed RNA Pol II ChIP and DRIP analyses. We observe a slight increase in R-loop and RNA Pol II accumulation in siDDX5 with pcDNA3 versus siCTL at RD (Fig 8D, middle and right panels). Flag-DDX5, but not Flag-DDX5-RK, decreased the R-loop formation at the poly(A) and RD with a decrease of RNA Pol II at RD in siDDX5 cells (Fig 8D). Moreover, both Flag-DDX5 WT and Flag-DDX5-RK restored the effect of siDDX5 on the R-loop accumulation at gene body (Fig 8D, right panel, RB), suggesting that the RGG/RG is required for R-loop suppression at the terminal region, but not within the gene body of β-actin. We next examined whether PRMT5-depleted cells also harbored an increase in RNA Pol II recruitment at RD of β-actin. Indeed, ChIP analysis showed an increased occupancy of RNA Pol II in the absence of PRMT5 at the poly(A) and RD (Fig 8E, middle panel). PRMT5 deficiency also increased R-loop accumulation at the β-actin gene locus Prom, RB, and RD (Fig 8E, right panel), consistent with the previous observation (Zhao et al, 2016). Importantly, RT–qPCR studies revealed that deficiency of DDX5 or XRN2 did not significantly

affect β-actin expression (Appendix Fig S9), suggesting that the increased accumulation of R-loop and RNA Pol II at transcriptional termination site was unlikely due to an increased transcription. PRMT5 knockdown decreased β-actin expression, suggesting that PRMT5 has multiple functions for gene expression and R-loop repression.

We proceeded with the detection of the occupancy of DDX5 and DDX5-RK at the β-actin gene locus. ChIP-qPCR analysis demonstrated that both Flag-DDX5 and Flag-DDX5-RK equally associated with the multiple regions of the β-actin gene locus (Fig 8F). These suggest that the DDX5 RGG/RG motif is not involved in targeting DDX5 to the chromatin per se, but plays a role in regulating association with interactors including XRN2.

## Discussion

In this study, we define a new function for DDX5 in R-loop resolution and reveal a regulatory role for the methylation of its RGG/RG motif in modulating its function. Depletion of either DDX5 or PRMT5 in cellulo leads to increased accumulation of RNA:DNA hybrids, as detected with the anti-S9.6 antibody by immunofluorescence, slot-blot analysis, DART method, and DRIP-qPCR. The C-terminal RGG/RG motif of DDX5 was methylated by PRMT5 and required for R-loop resolution, as amino acid substitution of its arginines in the RGG/RG motif with lysines (RK) failed to reverse the accumulation of R-loops in siDDX5 cells. Purified human DDX5 resolved R-loop structure in vitro, and its catalytically inactive mutant failed to rescue the effect of DDX5 deficiency on R-loop accumulation, suggesting that DDX5 represses R-loop accumulation by directly resolving the RNA:DNA hybrid structures. Moreover, proteomic analyses revealed that DDX5 associates with many RNA binding proteins including helicases, as well as with XRN2, an exonuclease known to function in transcriptional termination (Kim et al, 2004; West et al, 2004). This is in line with a few recent studies for high-throughput identification of R-loop-associated proteins, in which DDX5 was identified as a top hit of R-loop-associated proteins (Cristini et al, 2018; Wang et al, 2018), although the function of DDX5 was not addressed in these studies.

We show that one mechanism of action of DDX5 is to mediate regulated interaction with XRN2. DDX5-deficient cells harbored less XRN2 at the *β-actin* transcriptional termination site, accompanied by an increase in RNA Pol II accumulation. These data are consistent with pausing and R-loop accumulation and DDX5 functioning to recruit XRN2 at certain genomic regions in a PRMT5-dependent manner. It is known that p54nrb and PSF function to recruit XRN2 at transcription termination sites (Kaneko *et al*, 2007); therefore, DDX5 may function with these factors for optimal XRN2 recruitment. Consistent with DDX5 function in R-loop repression, DDX5 knockdown leads to spontaneous cellular DNA damage and hypersensitivity to replication stress.

Interestingly, compared to mammalian DDX5, Dbp2, the *S. cerevisiae* homolog of DDX5, lacks most of its C-terminal regulatory regions, but retains the RGG/RG motif. The fact that this motif is evolutionarily conserved further supports its importance for DDX5 function. RGG/RG motifs reside in flexible unfolded regions and mediate protein–protein interaction (Thandapani *et al*, 2013). We conclude that the DDX5 RGG/RG motif does not affect its *in vitro* R-loop resolution activity, i.e., helicase activity, nor the association of DDX5 with the chromatin, but rather, it prevents interaction with XRN2 for R-loop resolution at transcription termination pause sites of genes including *β-actin*, *HIST1H2BG, RPPH1, PRMT7, LINC01346, NFKBIL2, SLC25A3, JUN,* and *EEF1A1*. In addition, DDX5 has other R-loop resolving activities in gene bodies, for example, *EGR1* and *MALAT1* that function independent of XRN2 and it remains to be determined what protein(s) interact with the DDX5 RGG/RG motif for this function.

The methylation of the RGG/RG motif of DDX5 likely serves additional functions besides recruiting XRN2, and this is especially pertinent in R-loops that are not associated with transcriptional termination regions. Furthermore, the function we ascribe to the methylation of DDX5 probably functions in a coordinate fashion with the methylation of other substrates such as RNA Pol II subunit POLR2A to regulate R-loop suppression (Zhao *et al*, 2016). Besides DDX5, another DEAD-box helicase, DDX21 recently shown to regulate R-loops (Song *et al*, 2017), harbors an RGG/RG motif, and may be regulated in a similar manner as DDX5. As Tudor domains are known to interact with methylated arginines (Blanc & Richard, 2017), we do not know the protein interface that XRN2 uses to interact with methylated DDX5. Of note, XRN2 itself also contains an RGG/RG motif.

Why so many helicases are required to resolve R-loops? Certain RNA helicases may function on different types of R-loops or regulate R-loops for a particular process or at different genomic locations. For example, DDX1 converts G4 RNA structures to R-loops for IgH class switch recombination (Ribeiro de Almeida *et al*, 2018). DDX1 also functions in tRNA, mRNA and miRNA processing, transport of RNAs from nucleus to cytoplasm, and AU-rich element-mediated mRNA decay in cytoplasm. Its function in R-loop resolution facilitates homologous recombination repair (Li *et al*, 2016). DDX19 is well known for its functions in mRNA nuclear export at the nuclear pore. A role for DDX19 in R-loop suppression has been reported, suggesting a functional link between mRNA nuclear export and R-loop clearance (Hodroj *et al*, 2017). DDX23 is a component of the spliceosomal U5 small nuclear ribonucleoprotein (U5 snRNP) and required for integration of U4/U6 U5 tri-snRNP into the spliceosome. However, the role of DDX23 in suppressing R-loops does not require a functional U5 snRNP (Sridhara *et al*, 2017). DDX21 is

known to be involved in multiple steps of ribosome biogenesis as well as mRNA transcription (Song *et al*, 2017).

PRMT5-deficient cells exhibit induction of the p53 response, DNA damage, and cell death, and as such, PRMT5 is an interesting therapeutic target for many cancers (Chan-Penebre *et al*, 2015; Koh *et al*, 2015; Li *et al*, 2015; Kaushik *et al*, 2018). The upregulation of the p53 response was shown to be the result of an imbalance in the methylation of Sm proteins and other RNA binding proteins (e.g., SRSF) that regulate *MDM4* and *MDM2* alternative splicing (Bezzi *et al*, 2013; Dewaele *et al*, 2016; Gerhart *et al*, 2018). PRMT5-deficient cells also accumulate DNA damage leading to cellular senescence or cell death (Bezzi *et al*, 2013; Clarke *et al*, 2017; Hamard *et al*, 2018). What leads to the increased DNA damage in these cells is still not clear. Our observations herein suggest that PRMT5-defective cells fail to suppress R-loops leading to increased DNA damage. The increased R-loop accumulation in PRMT5-deficient cells may represent one of the earliest responses and may also contribute to some observed splicing defects. The DNA damage observed from the R-loop accumulation may also contribute to increasing the p53 response, as DNA damage is known to upregulate p53.

## Materials and Methods

### Reagents and antibodies

Mouse anti-DDX5 monoclonal antibodies (A-5, sc-166167 and clone204, 05-850) were purchased from Santa Cruz Biotechnology and Millipore. Rabbit anti-nucleolin antibody (ab50279) was from Abcam. Anti-XRN2 antibodies (A301-102A and Ab72181) were purchased from Bethyl Laboratories, Inc and Abcam (ab72181) for Western blot, and from Proteintech (11267-1-AP) for ChIP experiment. Mouse anti-γH2AX (05-636) and anti-signal stranded DNA (clone 16-19, MAB3034) monoclonal antibodies and rabbit anti-PRMT1 and anti-PRMT5 antibodies were obtained from Millipore. S9.6 antibody was a kind gift from Drs. Bedford and Yang (Yang *et al*, 2014) used for IF experiments and Western blot. For the DRIP experiments S9.6 was purified from the hybridoma purchased from the American Type Culture Collection (ATCC, Manassas, VA). Anti-nucleolin antibody (ab50279) was purchased from Abcam. Anti-RNA Pol II was from Santa Cruz Biotechnology, Inc (CTD4H8: sc-47701). Alexa Fluor-conjugated goat anti-rabbit antibodies and anti-mouse antibodies were from Invitrogen. *Escherichia coli* RNase H was purchased from New England Biolabs. Hydroxyurea, protein A-Sepharose, mouse anti-Flag, and α-tubulin monoclonal antibodies were from Sigma. Protease inhibitor cocktail and protein phosphatase inhibitor cocktail were from Roche. Monomethyl arginine (R*GG) (D5A12) rabbit monoclonal antibody and anti-symmetrical dimethylarginine antibody (SDMe-Arginine 13222S) were from Cell Signaling.

### Cell culture, siRNAs, plasmids, and transfection

All mammalian cells were cultured at 37°C with 5% $CO_2$. U2OS human osteosarcoma cells (ATCC), HEK293 cells (ATCC), and U2OS-TRE reporter cells were cultured in Dulbecco's modified Eagle's medium containing 10% v/v fetal bovine serum (FBS). U2OS cells were transfected with plasmid DNAs using

Lipofectamine 2000 or 3000 and siRNA oligonucleotides using Lipofectamine RNAiMAX (Invitrogen) according to the manufacturer's instructions. HEK293 cells were transfected with plasmid DNAs by standard calcium phosphate precipitation.

All siRNAs were purchased from Dharmacon. siRNA sequences are as follows: siDDX5 #1, 5′-ACA UAA AGC AAG UGA GCG AdTdT-3′; siDDX5 #2, 5′-CAC AAG AGG UGG AAA CAU AdTdT-3′; siDDX5 #3, 5′-CAA GUA GCU GCU GAA UAU UUU-3′; siXRN2, SMARTpool siGENOME human XRN2 siRNA (M-017622-01); siSenataxin, SMARTpool siGENOME human SETX siRNA (M-021420-01); siPRMT1, 5′-CGU CAA AGC CAA CAA GUU AUU-3′; siPRMT5, 5′-UGG CAC AAC UUC CGG ACU UUU-3′. The siRNA 5′-CGU ACG CGG AAU ACU UCG AdTdT-3′, targeting the firefly luciferase (GL2), was used as control. 20 nM siRNA was used for transfection. For co-transfection of two or more siRNAs, the total siRNA amount was adjusted to be the same in each sample by adding control siRNA (siLuc, GL2).

The N-terminal Flag-tagged DDX5 plasmid was constructed by inserting a Flag-coding sequence into the pcDNA3.1 (+) vector at the *Hind* III and *Bam* HI sites to get pcDNA3.1-Flag and then the PCR-amplified human DDX5 cDNA coding region at *Bam* HI and *Xho* I sites of pcDNA3.1-Flag vector. The Flag-DDX5 codon-silent mutant resistant to all three siDDX5 siRNAs used in this research was constructed in the pcDNA3.1-Flag vector using Gibson Assembly Cloning Kit (New England BioLabs, Inc.) according to the manufacturer's instructions. The gBlock DNAs were synthesized by Integrated DNA Technologies (IDT). The catalytic inactive DDX5 mutant with the replacement of both Arg403 at the motif Va and Arg428 at the motif VI with Leucine and the RK mutant with replacement of the five arginine residues with lysine at the RGG/RG motif were constructed by two-step PCR using the siRNA-resistant DDX5 construct as template. The plasmids for expressing GST fusion proteins of DDX5 fragments were constructed by inserting PCR-amplified DDX5 cDNA fragments in pGEX-6P1 vector at *Bam* HI and *Sal* I sites.

## Cell lysis and immunoprecipitation

For co-immunoprecipitation experiments, cells were lysed with a lysis buffer containing 50 mM HEPES, pH 7.4, 150 mM NaCl, 1% Triton X-100, and a cocktail of protease inhibitors and phosphatase inhibitors. After a brief sonication (five times 10 s) followed by high speed centrifugation, the supernatant was precleared and then incubated with antibodies for one hour and then protein A or protein G agarose beads for another hour at 4°C. In some experiments, the supernatant was incubated directly with agarose beads on which antibodies were pre-attached covalently. The beads were washed five times with lysis buffer for Western blot analysis or four time with lysis buffer and then twice with PBS for mass spectrometry analysis.

## SILAC (stable isotope labeling with amino acids in cell culture) and MS/MS mass analysis

U2OS stable cells with Flag-DDX5 expression were grown in standard heavy (Lys4/Arg6) SILAC medium and the control U2OS cells in light medium (Lys0/Arg0) for 10 days. Similarly, U2OS stable cells with Flag-DDX5 expression were grown in the heavy SILAC

medium and the control U2OS cells in light medium for 10 days. The cells were lysed and subjected to immunoprecipitation with anti-Flag antibody as described above. The beads with bound proteins were sent as described previously (Thandapani *et al*, 2013).

## Immunofluorescence

For immunostaining with S9.6, anti-Flag, and anti-nucleolin antibodies, the cells were fixed with cold methanol for 10 min at room temperature. Slides were then washed three times with PBS containing 0.1% Tween-20 (PBST) and blocked with blocking buffer (3% bovine serum albumin in PBST) for 1 h at room temperature or overnight at 4°C. Slides were incubated with antibodies S9.6 (1/200), anti-Flag (1/200), and anti-nucleolin (1/1,000) diluted in the blocking buffer for 2 h. After three washes with PBST, slides were incubated with corresponding fluorescent secondary antibodies for 2 h at room temperature. Slides were then washed three times with PBST before mounting with IMMU-MOUNT (Thermo Scientific) mounting medium containing 1 μg/ml of 4′,6-diamidino-2-phenylindole (DAPI). For immunostaining with anti-γ-H2AX antibody, cells were fixed for 10 min with 4% paraformaldehyde (PFA). After three washes with PBS, the cells were permeabilized for 5 min with 0.5% Triton X-100 in PBS. Coverslips were incubated overnight in PBS blocking buffer containing 10% FBS and 0.1% Triton X-100, and then incubated with primary antibody against γ-H2AX antibody (1:2,000) diluted in PBS containing 5% FBS for 30 min. After three washes, the coverslips were incubated with corresponding fluorescent secondary antibodies for another 30 min in PBS buffer containing 5% FBS. After rinsing, the coverslips were mounted with IMMU-MOUNT (Thermo Scientific) mounting medium containing 1 μg/ml of 4′,6-diamidino-2-phenylindole (DAPI). Images were taken using a Zeiss M1 fluorescence microscope with 63× amplification and Zeiss LSM800 confocal system with 40× magnification where indicated.

## DART assay

U2OS-TRE cells were first transfected with siRNAs and 24 h after siRNA transfection with plasmids expressing KillerRed (KR) (pBroad3 TA-KR and pBroad3 tetR-KR). 36–48 h after plasmids transfection, cells were exposed for 25 min to a 15W Sylvania cool white fluorescent lamp for ROS-induced damage through light-induced KR activation and let for 1 h to recover before fixation to start the S9.6 IF protocol. Cells were rinsed with PBS and fixed in 4% PFA for 15 min at room temperature. They were washed three times with PBS, permeabilized by 0.2% Triton X-100 in PBS for 10 min, and then washed three times with PBS. After that, cells were incubated in buffer (10 mM Tris–HCl, 2 mM EDTA, pH = 9) and heated on a 95°C heating block for 20 min to expose the antigen. The dish was cooled down and washed three times with PBS, and cells were blocked using 5% BSA in 0.1% PBST for 30 min at room temperature. The first and secondary antibodies were diluted in the same blocking buffer (anti-S9.6 1:500 and anti-mouse Alexa Fluor 488 1:1,000). The primary antibody was incubated for 2 h at room temperature, and then, cells were washed three times with PBS and incubated for 1 hr with the secondary antibody, following three more washes with PBS and incubation with DAPI. Three independent experiments were done, and 50 cells were analyzed in each

experiment. S9.6 intensity in the KillerRed foci area was quantified using Volocity software.

## Slot blotting

Nucleic acids were extracted from U2OS cells by SDS/proteinase K treatment at 37°C overnight followed by phenol–chloroform extraction and ethanol precipitation. The nucleic acids were blotted onto Hybond-N nylon membrane in duplicate using a slot-blot apparatus (Schleicher & Schuell). One half of the membrane was treated with 0.5 N NaOH and 1.5 M NaCl for 10 min to denature the DNA and neutralized for another 10 min in 0.5 M Tris–HCl buffer (pH 7.0) containing 1 M NaCl. After UV-cross-linking ($0.12 \text{ J/m}^2$), the non-treated membrane was subjected to Western blot analysis with S9.6, and the treated membrane, served as loading control, with the single-stranded DNA antibody. The S9.6 signal was normalized by the loading control.

## DRIP (RNA:DNA immunoprecipitation)-qPCR

DRIP assays were performed as described (Ginno *et al*, 2012). Briefly, nucleic acids were extracted from U2OS cells by SDS/proteinase K treatment at 37°C overnight followed by phenol–chloroform extraction using MaXtract™ High Density ($100 \times 15 \text{ ml}$ from Qiagen) and ethanol precipitation at room temperature. The harvested nucleic acids were digested for 24 h at 37°C using a restriction enzyme cocktail (50 units/100 μg nucleic acids, each of *Bsr*GI, *Eco*RI, *Hind*III, *Ssp*I, and *Xba*I) in the New England Biolabs CutSmart buffer with 2 mM Spermidine and 1X BSA. Digested DNAs were cleaned up by phenol–chloroform extraction using MaXtract™ High Density ($200 \times 2 \text{ ml}$) followed by treatment or not with RNase H (20 units/100 μg nucleic acids) overnight at 37°C in the New England Biolabs RNase H buffer. RNA:DNA hybrids from 4 μg digested nucleic acids, treated or not with RNase H, were immuno-precipitated using 10 μg of S9.6 antibody and 50 μl of protein A/G agarose beads at 4°C for 2 h or overnight in IP buffer (10 mM $NaPO_4$, 140 mM NaCl, 0.05% Triton X-100). The beads were then washed four times with IP buffer for 10 min at room temperature, and the nucleic acids were eluted with elution buffer (50 mM Tris–HCl, pH8.0, 10 mM EDTA, 0.5% SDS, and 70 μg of protease K) at 55°C for 1 h. Immunoprecipitated DNA was then cleaned up by a phenol–chloroform extraction followed by ethanol precipitation at −20°C for 1 h. Quantitative PCR was performed at the indicated regions using the primers listed in Appendix Table S1. Enrichment of RNA:DNA hybrids is calculated as percentage of input.

## ChIP (chromatin immunoprecipitation)

HEK293 cells were cross-linked with 1% formaldehyde at room temperature for 10 min. The reaction was stopped by adding glycine to a final concentration of 0.125 M. Then, the cells were washed twice with ice-cold PBS and then frozen at −80°C. ChIP experiment was proceeded using the SimpleChIP® Plus Sonication ChIP Kit (#56383) according to the manufacturer's protocol. Briefly, pellets corresponding to $4 \times 10^6$ cells were resuspended in 1× Chip sonication lysis buffer in the presence of protease inhibitors (PIC) and soni-cated to achieve a chromatin sized of 300–1,000 bp using a Branson 450 CE Sonicators (total 4 min run, 1 s ON and 1 s OFF) on ice. After

pelleting debris, the equivalent 10 μg of sonicated, cross-linked chromatin was incubated overnight with desired antibody and then incu-bated 2 h with protein G magnet beads. The beads were washed with furnished buffer, as indicated by the protocol kit. Chromatin was eluted from the beads with elution buffer at 65°C. The cross-linking of the eluted chromatin as well as the input was treated for 30 min with RNase A at 37°C in the presence of 200 mM of NaCl, and then reverse-cross-linked at 65°C for 2 h in the presence of proteinase K. The chromatin was purified by column purification kit and eluted in 50 μl of elution buffer. The enriched chromatin was analyzed by qPCR using primers listed in Table S1. For ChIP Pol II results, the relative signal was normalized to the IgG, and for ChIP XRN2 results, the signal was normalized to the Pol II.

## *In vitro* methylation assay

GST-tagged DDX5 fragments and mutants were purified from bacteria. 10 μg of each GST-tagged construct was incubated with 2 μl of (methyl-$^3$H) S-adenosyl-L-methionine solution (15 Ci/mmol stock solution, 0.55 μM final concentration, PerkinElmer) and 2 μg of PRMT5:MEP50 active complex (Sigma-Aldrich) in methylation buffer (HEPES, pH 7.4 50 mM, NP-40 0.01%, DTT 1 mM, PMSF 1 mM) for 2 h at 37°C. Samples were separated by SDS–PAGE and stained with Coomassie Blue. After de-staining, the gel was then incubated for 1 h in EN$^3$HANCE (PerkinElmer) followed by 30-min wash in cold water, according to the manufacturer's instructions and the reaction was visualized by fluorography.

## FACS-based cell survival assay

For FACS-based cell survival assay, a U2OS cell line with stably transfected GFP (U2OS-GFP) was generated. The U2OS and U2OS-GFP cells were transfected with control and target siRNAs respec-tively, or reversely, the U2OS cells were transfected with target siRNAs and U2OS-GFP cells with control siRNA. The cells were tryp-sinized and mixed with an approximately 1:1 ratio 2 days after transfection. The cells were then co-plated and treated with DNA damage agents or left untreated. After 7–10 days of recovery, the cells were subjected to FACS analysis to determine the ratio of $GFP^+/GFP^-$ (green/non-green) cells, which reflects the relative survival of the two cell populations.

## *In vitro* unwinding assay

DDX5 and XRN2 were tagged at the N-terminus with GST and at the C-terminus with $His_{10}$, expressed in bacteria or Sf9 insect cells, as indicated, and purified as described for PALB2 (Buisson *et al*, 2014). R-loop and D-loop substrates were generated by annealing purified oligonucleotides: DNA strand 1: 5′-GGGTGAACCTG-CAGGTGGGCGGCTGCTCATCGTAGGTTAGTTGGTAGAATTCGGCA GCGTC-3′ and DNA strand 2: 5′-GACGCTGCCGAATTCTACC AGTGCCTTGCTAGGACA TCTTTGCCCACCTGCAGGTTCACCC-3′ with either RNA strand: 5′-AAAGAUGUCCUAGCAAGGCAC-3′ (or DNA strand: 5′-AAAGATGTCCTAGCAAGGCAC-3′, or 5′ protuberant RNA strand: 5′ ACUCACUCACUCAAAAGAUGUCCUAGCAAGGCAC-3′. Unwinding assays were performed in MOPS buffer (25 mM MOPS (morpholinepropanesulfonic acid), pH 7.0, 60 mM KCl, 0.2% Tween-20, 2 mM DTT, 5 mM ATP, 5 mM $MgCl_2$). DDX5 and

labeled R-loop or D-loop (100 nM) substrates were incubated in MOPS buffer for 20 min at 37°C, followed by deproteinization in one-fifth volume of stop buffer (20 mM Tris-Cl, pH 7.5, and 2 mg/ml proteinase K) for 20 min at 37°C. Reactions were loaded on an 8% acrylamide gel, electrophoresed at 150V for 120 min, dried onto filter paper, and autoradiographed.

### R-loop RNase assays

XRN2 RNase assays were performed in Tris/MOPS buffer (12.5 mM MOPS (morpholinepropanesulfonic acid), pH 7.0, 25 mM Tris–HCl, pH 7.9, 30 mM KCl, 50 mM NaCl, 0.1% Tween-20, 1.5 mM DTT, 5 mM ATP, 10 mM MgCl$_2$). DDX5 and labeled R-loop (100 nM) substrates were incubated in Tris/MOPS buffer for 20 min at 37°C, and XRN2 was added for 20 min at 37°C. Reactions were deproteinized in one-fifth volume of stop buffer (20 mM Tris-Cl, pH 7.5, and 2 mg/ml proteinase K) for 20 min at 37°C. Reactions were loaded on an 8% acrylamide gel, electrophoresed at 150V for 120 min, dried onto filter paper, and autoradiographed.

### Clonogenic cell survival assay

For clonogenic assay, 200–1,000 cells/10-cm dish were seeded and the cells were treated with hydroxyurea (HU) for 20 h. Ten to Fourteen days after the treatment, cells were fixed with 4% paraformaldehyde and stained with 0.05% crystal violet (Sigma-Aldrich) and colonies were counted.

### Reverse transcription and real-time quantitative PCR (RT–qPCR) analysis

Total cellular RNA was isolated from siRNA-transfected U2OS cells using GenElute$^{TM}$ Mammalian Total RNA Miniprep Kit (Sigma) according to the manufacturer's instructions. For each experimental condition, three independent siRNA transfection experiments were carried out for the isolation of RNA. cDNA was synthetized using Promega$^{TM}$ M-MLV Reverse Transcription Kit and analyzed by real-time quantitative PCR (RT–PCR) using SYBR Green PCR Master Mix (Applied Biosystems). The primers used are given in Appendix Table S1. *Gapdh* housekeeping gene expression was used to normalize gene expression.

**Expanded View** for this article is available online.

## Acknowledgements

We thank Drs. Mark Bedford and Yanzhong Yang for purified S9.6 antibody and helpful discussions. We also thank Haibo Yang and Li Lan (MGH) for DART assay reagents and advice, and Marie-Christine Caron for expert technical assistance. This work was funded by FDN-154303 to S.R. and FDN-388879 to J.Y.M. M.K. is a recipient of a Doctoral Cole Foundation Fellowship. F.F.B. received a scholarship from the Emerging Leaders in the Americas Program (ELAP). J.Y.M. is a FRQS chair in genome stability.

## Author contributions

ZY, SYM, J-YM, and SR designed the research; ZY, SYM, YC, MK, and FFB performed the experiments; ZY, SYM, MK. J-YM, and SR analyzed the data; and ZY, J-YM, and SR wrote the paper.

## Conflict of interest

The authors declare that they have no conflict of interest.

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
