## [Review Process File · The EMBO Journal]

Arginine methylation of the DDX5 helicase RGG/RG motif by PRMT5 regulates resolution of RNA:DNA hybrids

Sofiane Y. Mersaoui, Zhenbao Yu, Yan Coulombe, Martin Karam, Franciele F. Busatto, Jean-Yves Masson and Stéphane Richard.

Review timeline:

Submission date:	23 rd October 2018
Editorial Decision:	22 nd November 2018
Revision received:	8 th April 2019
Editorial Decision:	25 th April 2019
Revision received:	15 th May 2019
Accepted:	28 th May 2019

Editor: Hartmut Vodermaier

Transaction Report:

1st Editorial Decision

22nd November 2018

Thank you for submitting your manuscript on arginine methylation of DDX5 in regulation of RNA:DNA hybrid structures. We have now received the reports of three expert referees, copied below for your information. As you will see, all reviewers acknowledge the general interest of this work as well as the potential importance of your new findings. At the same time, especially reviewers 1 and 3 raise several substantive concerns that would need to be adequately addressed before the study may become suitable for acceptance. One shared major issue is that the study would need to be extended beyond the sole example of the beta-actin gene, ideally using genome-wide approaches such as DRIP-seq. Both critical reviewers also share concerns related to the interplay with XRN2 and transcription vs termination effects.

Should you be able to satisfactorily address these key points as well as the various other, more specific experimental and presentational issues noted by all three referees, we shall be happy to consider a revised manuscript further for publication in The EMBO Journal. Please remember that our policy to allow only a single round of major revision makes it important to carefully answer to all points raised at this stage - therefore, please do not hesitate to get back to me with any questions/comments you may have regarding the referee reports already during the early stages of your revision. We might also discuss possible extension of the revision period (beyond the regular three months), during which time the publication of any competing work elsewhere would have no negative impact on our final assessment of your own study.

REFeree REPORTS

Referee #1:

The paper by Mersaoui et al describes novel post-translational modification, arginine methylation, on helicase DDX5 which is involved in R-loop resolution. In particular, the authors show that

DDX5 is an RNA/DNA helicase in vitro and its knock-down results in accumulation of R-loops using DRIP, slot blot and S9.6 immuno-fluorescence. The arginine methylation on DHX9 is mediated by PRMT5 and it is required for interaction between DHX9 and XRN2, previously implicated in the process of transcriptional termination. The authors demonstrate that DDX5 knock-down results in transcriptional read-through and reduction in XRN2 recruitment on b-actin gene.

Overall speaking this ms presents an interesting set of data demonstrating a novel function of arginine methylation which regulates the function of R-loop resolving factor, DDX5. Having said that, experiments presented in this ms require additional mechanistic insights and controls to support the claims made by the authors.

Major comments:

1. The mechanistic aspects of DDX5 function on transcriptional termination are not fully investigated. The effect of DDX9 kd on transcriptional termination was demonstrated just for one gene, b-actin. The authors are making strong conclusions about molecular mechanisms based on this one example, but how general these conclusions are? Does it happen with all the genes genome-wide, or is it a very gene-specific phenomenon?
2. DDX5 KD results in decrease of transcription over the TSS and gene body of b-actin gene (based on Pol II CHIP data). Therefore, the defect in transcriptional termination is maybe just a consequence of reduced transcription and not due to disrupted interaction with Xrn2. Does DDX5 mutant in RGG domain have the same effect on transcriptional termination as DDX5 knock-down? What is the binding profile of the WT DDX5 - does it bind termination regions together with Xrn2? Does DDX5 RGG mutant still bind termination regions?
3. Some of the IF images presented throughout the paper do not seem to correspond to the conclusions drawn by the authors. In particular, Figure 1D claims an increase in a number of Rad51 foci, however this is not clear from the Image (no clear foci are seen!). Why the cells are not presented at the same magnification (and clarity) as in panel 1B?
4. The paper claims that there was no increase in g-H2AX foci (Fig. S1E), however the overall levels of g-H2AX are increased by WB (Figure S4A). What is the explanation to this apparent discrepancy?
5. The authors demonstrate that either helicase mutation or mutation in RGG motif of DDX5 are important for R-loop resolution using S9.6 IF. Is the same effect observed using DRIP analysis on the genes presented in Figure 2D?

Minor comment:

1. Representative images of raw data for S9.6 IF, corresponding to Figures 2C, 3E, 4E and F should be presented in supplementary materials.

Referee #2:

The Merasoui et al manuscript "Arginine methylation of DDX5 RGG/RG motif by PRMT5 regulates RNA:DNA resolution" identifies PRMT5-mediated methylation of the helicase DDX5 facilitates R-loop resolution to prevent DNA damage. Overall this is important and novel work that will be suitable for publication in EMBO J if two comments are addressed.

Specific comments:

- 1) It is important to test whether DDX5 arginine methylation levels change upon depletion of PRMT5 by RNAi or CRISPR/Cas9.
- 2) The following line in the abstract could be phrased differently: "PRMT5, an arginine methyltransferase, associated with DDX5 and methylated its RGG/RG motif." For example: The

arginine methyltransferase PRMT5 associates with and methylates DDX5 at its RGG/RG motif"

Referee #3:

In the manuscript entitled "Arginine methylation of DDX5 RGG/RG motif by PRMT5 regulates RNA:DNA resolution" by Stéphane Richard and colleagues, the authors show that the DEAD-box helicase DDX5 can suppress or resolve R-loop structures in vitro and in vivo. Furthermore, DDX5 interacts with and is post-translationally modified by the arginine-methyltransferase PRMT5. Therefore, DDX5- as well as PRMT5-depleted cells show a substantial increase in the cellular level of these co-transcriptional structures, suggesting that this helicase and its posttranslational modification state play an important role in cellular R-loop resolution (Figure 2 and 3). The authors can also map an RGG/RG motif located at the C-terminus of DDX5 that is responsible for the regulation of this activity (Figure 4). Finally, the authors show by Co-IP's and mass spec that DDX5 interacts with the 5' to 3' exonuclease XRN2 and this interaction is mediated at the same region of DDX5 that is arginine methylated by PRMT5. This interaction appears to be important for efficient transcription termination at the beta-actin gene locus (Figure 5).

An intriguing question in this field is why cells have evolved so many different helicases with potentially redundant R-loop resolution activity. As discussed by the authors, an obvious hypothesis is that certain RNA helicases may function on different types of R-loops or regulate R-loops for a particular process or at different genomic locations but such a specificity has not been well established at the mechanistic level for SETX, AQR and most other DDX helicases. Therefore, based on the data provided, I think this study has great potential to go beyond previous studies and show at the mechanistic level that a new pathway that includes DDX5, PRMT5 and XRN2 has a role for R-loop resolution at specific functional regions of the genome (e.g. transcription termination sites). However, at the current state of the manuscript, this conclusion needs further experimental substantiation. If the authors can show convincingly that DDX5 has (such) a specificity towards a certain subset of R-loops (e.g. at transcription termination sites), this study would be appropriate for publication in EMBO.

MAJOR:

- In Figure 1, the authors show that IR-induced DSB repair is delayed in DDX5-depleted cells and this delay appears to be R-loop dependent. As the role of RNA:DNA hybrids during DSB repair is at best controversially discussed in the field at the moment (see Ohle et al., Cell 2016 and Zhao et al., EMBO Reports 2018), I believe that these data are rather a distraction from the major topic of this manuscript (the role of DDX5 in R-loop resolution at genic regions and not at DSB repair sites which could involve a completely different mechanism). Therefore, the authors should consider to remove the data in Figure 1 and S1, which can be used in a follow-up manuscript about a potential role of DDX5 at DSB repair sites.
- Figure 2C/D show increased R-loop accumulation in DDX5-depleted cells in the nucleus and at specific genomic loci. Are the primers used in the DRIP-qPCR experiments located at the promoter, gene body or transcription termination regions of these genes? The authors should test different primer pairs along the genes to analyze whether the difference of R-loop levels is more or less pronounced at specific functional elements of these genes (e.g. promoter, gene body and termination regions, see comment above).
- A substantial improvement of the manuscript would be to perform DRIP-Seq in siCTRL, siDDX5 and siPRMT5-depleted cells and map the location and changes of R-loop distribution genome-wide. This would give a much more comprehensive understanding about the role of DDX5 and PRMT5 in R-loop resolution at transcription termination sites. This conclusion is currently based only on a single gene, namely the actively transcribed beta-actin gene. In addition, ChIP-Seq of RNAPII under the same conditions would complement these data and provide a genome-wide view of transcriptional read-through upon DDX5 or PRMT5 depletion. As DRIP-qPCR and ChIP-qPCR appear to be well established in the lab, making libraries of these samples and sequence them should not be out of the scope for this study.
- Figure 5F shows that XRN2 degrades the RNA only when released by DDX5 in vitro. This is not surprising as XRN2 requires a free 5' RNA-end to allow 5' to 3' exonuclease activity which is not provided in the used R-loop substrate. The authors should address this issue by using a different RNA primer with an extended non-complementary 5' overhang that is not annealed to the DNA. This could give further mechanistic insights whether XRN2 alone can degrade the R-loop substrate even in the absence of DDX5.

MINOR:

- The BrdU foci experiment shown in Figure S1D should be more thoroughly described in the Materials and Methods section. I assume it's a native BrdU assay after pulse labelling with BrdU to visualize increased amounts of single-stranded DNA but there are no details about the BrdU labelling procedure provided.
- One open question is whether arginine methylation of DDX5 mediated by PRMT5 directly enhances the enzymatic R-loop resolution activity of DDX5. The authors should repeat the R-loop resolution assays and compare the activities of in vitro PRMT5-methylated DDX5 (Figure 4C/D) and unmethylated DDX5 on the in vitro R-loop substrates (Figure 2B).
- Are DDX5 and PRMT5's R-loop suppression/resolution activities (Figure 3E/F) related/work together in the same pathway? To address this, the authors should determine R-loop accumulation by S9.6 staining and DRIP-qPCR after double-knockdown of DDX5 and PRMT5.
- The efficiency of the knockdown conditions of XRN2 (Figure 5I) and PRMT5 (Figure 5J) are not shown and should be evaluated by Western blot and/or RT-qPCR analysis.
- In Figure 5, the authors should also assess the level of R-loops by DRIP-qPCR at the indicated primer locations of the beta-Actin gene in siCTRL, DDX5 and PRMT5 depleted cells to further investigate the interdependence of these factors and their RNA:DNA hybrid resolution activities.

1st Revision - authors' response

8th April 2019

Thank you for submitting your manuscript on arginine methylation of DDX5 in regulation of RNA:DNA hybrid structures. We have now received the reports of three expert referees, copied below for your information. As you will see, all reviewers acknowledged the general interest of this work as well as the potential importance of your new findings. At the same time, especially reviewers 1 and 3 raise several substantive concerns that would need to be adequately addressed before the study may become suitable for acceptance.

One shared major issue is that the study would need to be extended beyond the sole example of the beta-actin gene, ideally using genome-wide approaches such as DRIP-seq. Both critical reviewers also share concerns related to the interplay with XRN2 and transcription vs termination effects.

REPLY: Indeed this is an important question that we address extensively in this revised version. Of the six loci in the initial version, we searched the R-loops position within the gene of interest ie at the promoter region, gene body, or transcriptional termination site. For *EGR1* and *MALAT1*, R loop database revealed that once digested with the restriction enzymes of the DRIP protocol the R-loops we amplified (see Figure 1D) resided within the gene body, while R-loops of *HIST1H2BG* and *RPPH1* extended to the transcription termination region. Actually, *HIST1H1E* and *NEAT1* had complicated overlapping R-loops. *HIST1H1E*'s R-loop corresponds to its transcription termination site of *HIST1H1E* and the promoter and gene body of a neighbouring gene (*HIST1H2BD*). As for *NEAT1*, it encodes two isoforms and the R-loop fragment is at the transcription termination of the short isoform, but in the gene body of the long isoform. For these reasons, we did not pursue analysis with these 2 loci.

Reviewer Figure 1. The gene location and genomic qPCR amplification region are shown at the top of each panel. B, E, H, S and X denote the location of the *BsrGI*, *EcoRI*, *HindIII*, *SspI*, and *XbaI*. The RNA transcripts are shown. The identified R-loop peaks were extracted from the R-loop database (R-loop DB) for each region.

Furthermore, we address this issue of the coordinated role of DDX5 and XRN2 at transcription termination site using RNA pol II ChIP-qPCR and DRIP-qPCR data at 6 new loci (new Figure 7). In sum, our data show that DDX5, like XRN2, is required for R-loop resolution at transcription termination sites to facilitate RNA Pol II release of certain loci including *HIST1H2BG*, *RPPH1*, *PRMT7*, *LINC01346*, *NFKBIL2*, *SLC25A3*, *JUN*, *EEF1A1* and *b-actin*.

In addition we confirm the accumulation of R-loops in the absence of DDX5 and PRMT5 using the DART (DNA damage at RNA transcription) system (Teng et al., 2018; Liang et al, 2019).

Referee #1:

The paper by Mersaoui et al describes novel post-translational modification, arginine methylation, on helicase DDX5 which is involved in R-loop resolution. In particular, the authors show that DDX5 is an RNA/DNA helicase *in vitro* and its knock-down results in accumulation of R-loops using DRIP, slot blot and S9.6 immuno-fluorescence. The arginine methylation on DHX9 is mediated by PRMT5 and it is required for interaction between DHX9 and XRN2, previously implicated in the process of transcriptional termination. The authors demonstrate that DDX5 knock-down results in transcriptional read-through and reduction in XRN2 recruitment on b-actin gene.

Overall speaking this ms presents an interesting set of data demonstrating a novel function of arginine methylation which regulates the function of R-loop resolving factor, DDX5. Having said that, experiments presented in this ms require additional mechanistic insights and controls to support the claims made by the authors.

Major comments:

1. The mechanistic aspects of DDX5 function on transcriptional termination are not fully investigated. The effect of DDX5 kd on transcriptional termination was demonstrated just for one gene, b-actin. The authors are making strong conclusions about molecular mechanisms based on this one example, but how general these conclusions are? Does it happen with all the genes genome-wide, or is it a very gene-specific phenomenon?

REPLY: This is an important point and we performed extensive experimentation in this revised version to address the mechanism of action of the RGG/RG motif of DDX5 and its methylation by PRMT5. It is indeed well-documented that XRN2 functions mainly at transcription termination regions of genes (Kim et al., 2004; West et al., 2004; Morales et al., 2016). We used 4 loci (*EGRI*, *MALAT1*, *HIST1H2BG*, and *RPPH1* loci) and XRN2 knockdown caused R-loop accumulation at two of the four loci (Figure 6E; *HIST1H2BG*, and *RPPH1*). Interestingly, the DNA fragments we amplified at each loci, between indicated restriction sites, corresponded to areas of R-loops extending over the termination pause sites for *HIST1H2BG*, and *RPPH1*. The amplified fragments for *EGRI* and *MALAT1* fragments were in gene bodies and were not significantly enriched in the absence of XRN2 by DRIP. These results suggest that DDX5 functions with XRN2 to resolve R-loops at transcription termination sites, but also has additional XRN2 independent functions in R-loop suppression. To further address whether DDX5 and XRN2 function at transcription termination sites, we analyzed 6 other loci (*PRMT7*, *LINC01346*, *NFKBIL2*, *SLC25A3*, *JUN*, *EEF1A1*). Three loci (*PRMT7*, *LINC01346*, *NFKBIL2*) were selected due to their known R-loop formation at the transcriptional terminal regions from the R-loop database (Wongsurawat et al., 2012) and three loci were selected because they are known to be regulated by XRN2 (*SLC25A3*, *JUN*, *EEF1A1*) (Fong et al., 2015). In all cases examined R-loops were detected by S9.6 DRIP-qPCR in either XRN2 or DDX5 depleted cells (Figure 7). RNA Pol II accumulated at transcription termination pause sites of these 6 loci in XRN2 or DDX5 deficient cells (Figure 7). These findings demonstrate that DDX5, like XRN2, is required for R-loop resolution at transcription termination sites to facilitate RNA Pol II release of certain loci.

We then examined the role of RGG/RG motif at the b-actin poly (A) and region D (downstream of the poly(A), a region known to form R-loops and accumulate RNA Pol II (Kaneko et al., 2007 and Skourti-Stathaki et al, 2011). Flag-DDX5, but not Flag-DDX5-RK, decreased the R-loop formation at the poly(A) and region D with a decrease of RNA Pol II at the region D in siDDX5 cells (Figure 8D). ChIP-qPCR analysis demonstrated that both Flag-DDX5 and Flag-DDX5-RK equally associated with the multiple regions of the *β-actin* gene locus (Figure 8F). These findings suggest that the DDX5 RGG/RG motif is not involved in targeting DDX5 to the chromatin *per se*, but plays a role in regulating association with interactors including XRN2.

In sum, although we do not provide genome-wide data, we focus on defining the mechanism of action of DDX5. We conclude that the DDX5 RGG/RG motif does not affect its *in vitro* R-loop resolution activity i.e. helicase activity nor the association of DDX5 with the chromatin, but rather it is involved in mediating an interaction with XRN2 for R-loop resolution at transcription termination pause sites of genes including *b-actin*, *HIST1H2BG*, *RPPH1*, *PRMT7*, *LINC01346*, *NFKBIL2*, *SLC25A3*, *JUN*, and *EEF1A1*. In addition, DDX5 has other R-loop resolving activities in gene bodies, for example, that function independent of XRN-2.

2. DDX5 KD results in decrease of transcription over the TSS and gene body of b-actin gene (based on Pol II ChIP data). Therefore, the defect in transcriptional termination is maybe just a consequence of reduced transcription and not due to disrupted interaction with Xrn2. Does DDX5 mutant in RGG domain have the same effect on transcriptional termination as DDX5 knock-down? What is the binding profile of the WT DDX5 - does it bind termination regions together with Xrn2? Does DDX5 RGG mutant still bind termination regions?

REPLY: Indeed transcription is required to generate the RNAs that form the R-loops. We measured the mRNA levels of *EGRI*, *MALAT1*, *HIST1H2BG*, *RPPH1* and b-actin in U2OS cells with siDDX5, siXRN2 and siPRMT5 and we did not observe a significant increase that correlate with R-loops formation, except for *MALAT1* and b-actin with siPRMT5 (Figure S9), consistent with its role in regulating gene expression.

We examined the role of RGG/RG motif at the b-actin poly (A) and region D (downstream of the poly(A), a region known to form R-loops and accumulate RNA Pol II (Kaneko et al., 2007 and Skourti-Stathaki et al, 2011). Flag-DDX5, but not Flag-DDX5-RK, decreased the R-loop formation at the poly(A) and region D with a decrease of RNA Pol II at the region D in siDDX5 cells (Figure 8D) suggesting that the RGG/RG motif is required for R-loop resolution and RNA Pol II displacement from the termination zone. ChIP-qPCR analysis demonstrated that both Flag-DDX5 and Flag-DDX5-RK equally associated with the multiple regions of the β -actin gene locus (Figure 8F). These suggest that the DDX5 RGG/RG motif is not involved in targeting DDX5 to the chromatin *per se*, but plays a role in regulating association with interactors including XRN2.

We conclude that the DDX5 RGG/RG motif does not affect its *in vitro* R-loop resolution activity i.e. helicase activity (Figure S5) nor the association of DDX5 with the chromatin (Figure 8F), but rather it is involved in interacting with XRN2 for R-loop resolution at transcription termination pause sites of genes including *b-actin*, *HIST1H2BG*, *RPPH1*, *PRMT7*, *LINC01346*, *NFKBIL2*, *SLC25A3*, *JUN*, and *EEF1A1* (Figure 6E, 7).

3. Some of the IF images presented throughout the paper do not seem to correspond to the conclusions drawn by the authors. In particular, Figure 1D claims an increase in a number of Rad51 foci, however this is not clear from the Image (no clear foci are seen!). Why the cells are not presented at the same magnification (and clarity) as in panel 1B?

REPLY: As recommended by reviewer #3, we have deleted the RAD51 foci, as they provide a distraction from the main message of the manuscript.

4. The paper claims that there was no increase in g-H2AX foci (Fig. S1E), however the overall levels of g-H2AX are increased by WB (Figure S4A). What is the explanation to this apparent discrepancy?

REPLY: We apologize for the confusion. The cells in Fig. S1E were treated with 10 Gy ionizing radiation (IR) cells and those of Fig S4A, S4B were left untreated to examine spontaneous DNA damage foci. In Fig. S1E, both siCTL and siDDX5 had maximal amounts of DNA damage (g-H2AX) foci with little difference between the number of foci between the samples, as noted by the reviewer. This shows that IR induces maximal amount of DNA damage in either siCTL or siDDX5 cells. In contrast, Fig. S4A shows the spontaneous damage (without DNA damage) observed in siCTL and siDDX5. DDX5 deficiency caused an increase of spontaneous DNA damage (Figure S4A, S4B, the γ H2AX-positive cells increase from 4% to 14% now Figure S2A, S2B).

In sum, as recommended by reviewer #3, we deleted the RAD51 foci along with the g-H2AX foci of Fig. S1E in the revised version. Reviewer 3 comment (see below) "I believe that these data are rather a distraction from the major topic of this manuscript (the role of DDX5 in R-loop resolution at genic regions and not at DSB repair sites which could involve a completely different mechanism)"

5. The authors demonstrate that either helicase mutation or mutation in RGG motif of DDX5 are important for R-loop resolution using S9.6 IF. Is the same effect observed using DRIP analysis on the genes presented in Figure 2D?

REPLY: This is an important experiment for defining the mechanism of action of the RGG motif.

We performed a rescue experiment in DDX5-deficient cells with either Flag-DDX5 or Flag-DDX5-RK and we examined R-loops using DRIP-qPCR analysis (Figure 5H). As expected, the WT DDX5 restored the effect of siDDX5 siRNA on R-loop accumulation at six loci analyzed (Figure 5H). By contrast, the DDX5-RK mutant could not rescue the effect of siDDX5 (*EGRI*, *MALATI*, *HIST1H2BG*, or *RPPH1*, Figure 5H). These results suggest that the RGG/RG motif is required for the regulation of DDX5 function in cellular R-loop resolution.

Minor comment:

1. Representative images of raw data for S9.6 IF, corresponding to Figures 2C, 3E, 4E and F should be presented in supplementary materials.

REPLY: As suggested, we added representative images for S9.6 IF in Figure S1A (for Figure 1C), Figure 2F (for Figure 2F), Figure S4 (for Figure 5F and 5G) and S8C (for Figure S8C).

Referee #2:

The Merasoui et al manuscript "Arginine methylation of DDX5 RGG/RG motif by PRMT5 regulates RNA:DNA resolution" identifies PRMT5-mediated methylation of the helicase DDX5 facilitates R-loop resolution to prevent DNA damage. Overall this is important and novel work that will be suitable for publication in EMBO J if two comments are addressed.

Specific comments:

1) It is important to test whether DDX5 arginine methylation levels change upon depletion of PRMT5 by RNAi or CRISPR/Cas9.

REPLY: This is an important experiment. We now add new data showing that DDX5 is hypomethylated in cells transfected with siRNA targeting PRMT5, but not siControl (new Figure 5A).

2) The following line in the abstract could be phrased differently: "PRMT5, an arginine methyltransferase, associated with DDX5 and methylated its RGG/RG motif." For example: The arginine methyltransferase PRMT5 associates with and methylates DDX5 at its RGG/RG motif"

REPLY: The sentence was edited as suggested by the reviewer.

Referee #3:

In the manuscript entitled "Arginine methylation of DDX5 RGG/RG motif by PRMT5 regulates RNA:DNA resolution" by Stéphane Richard and colleagues, the authors show that the DEAD-box helicase DDX5 can suppress or resolve R-loop structures in vitro and in vivo. Furthermore, DDX5 interacts with and is post-translationally modified by the arginine-methyltransferase PRMT5. Therefore, DDX5- as well as PRMT5-depleted cells show a substantial increase in the cellular level of these co-transcriptional structures, suggesting that this helicase and its posttranslational modification state play an important role in cellular R-loop resolution (Figure 2 and 3). The authors can also map an RGG/RG motif located at the C-terminus of DDX5 that is responsible for the regulation of this activity (Figure 4). Finally, the authors show by Co-IP's and mass spec that DDX5 interacts with the 5' to 3' exonuclease XRN2 and this interaction is mediated at the same region of DDX5 that is arginine methylated by PRMT5. This interaction appears to be important for efficient transcription termination at the beta-actin gene locus (Figure 5).

An intriguing question in this field is why cells have evolved so many different helicases with potentially redundant R-loop resolution activity. As discussed by the authors, an obvious hypothesis is that certain RNA helicases may function on different types of R-loops or regulate R-loops for a particular process or at different genomic locations but such a specificity has not been well established at the mechanistic level for SETX, AQR and most other DDX helicases. Therefore, based on the data provided, I think this study has great potential to go beyond previous studies and show at the mechanistic level that a new pathway that includes DDX5, PRMT5 and XRN2 has a role for R-loop resolution at specific functional regions of the genome (e.g. transcription termination sites). However, at the current state of the manuscript, this conclusion needs further experimental substantiation. If the authors can show convincingly that DDX5 has (such) a specificity towards a certain subset of R-loops (e.g. at transcription termination sites), this study would be appropriate for publication in EMBO.

REPLY: This is an important point and we performed extensive experimentation in this revised version to address the mechanism of action of the RGG/RG motif of DDX5 and its methylation by PRMT5. It is indeed well documented that XRN2 functions mainly at transcription termination regions of genes (Kim et al., 2004; West et al., 2004; Morales et al., 2016). We used 4 loci (*EGR1*, *MALAT1*, *HIST1H2BG*, and *RPPH1* loci) and XRN2 knockdown caused R-loop accumulation at two of the four loci (Figure 6E; *HIST1H2BG*, and *RPPH1*). Interestingly, the DNA fragments we amplified at each loci, between indicated restriction sites, corresponded to areas of R-loops extending over the termination pause sites for *HIST1H2BG*, and *RPPH1*. The amplified fragments for *EGR1* and *MALAT1* fragments were in gene bodies and were not significantly enriched in the absence of XRN2 by DRIP. These results suggest that DDX5 functions with XRN2 to resolve R-loops at transcription termination sites, but also has additional XRN2 independent functions in R-loop suppression. To further address whether DDX5 and XRN2 function at transcription termination sites, we analyzed 6 other loci (*PRMT7*, *LINC01346*, *NFKBIL2*, *SLC25A3*, *JUN*, *EEF1A1*). Three loci (*PRMT7*, *LINC01346*, *NFKBIL2*) were selected due to their known R-loop formation at the transcriptional terminal regions from the R-loop database (Wongsurawat et al., 2012) and three loci were selected because they are known to be regulated by XRN2 (*SLC25A3*, *JUN*, *EEF1A1*) (Fong et al., 2015). In all cases examined R-loops were detected by S9.6 DRIP-qPCR in either XRN2 or DDX5 depleted cells (Figure 7). RNA Pol II accumulated at transcription termination pause sites of these 6 loci in XRN2 or DDX5 deficient cells (Figure 7). These findings demonstrate that DDX5, like XRN2, is required for R-loop resolution at transcription termination sites to facilitate RNA Pol II release of certain loci.

MAJOR:

- In Figure 1, the authors show that IR-induced DSB repair is delayed in DDX5-depleted cells and this delay appears to be R-loop dependent. As the role of RNA:DNA hybrids during DSB repair is at best controversially discussed in the field at the moment (see Ohle et al., Cell 2016 and Zhao et al., EMBO Reports 2018), I believe that these data are rather a distraction from the major topic of this manuscript (the role of DDX5 in R-loop resolution at genic regions and not at DSB repair sites which could involve a completely different mechanism). Therefore, the authors should consider to remove the data in Figure 1 and S1, which can be used in a follow-up manuscript about a potential role of DDX5 at DSB repair sites.

REPLY: We agree with the reviewer. Figure 1 and S1 were deleted.

• Figure 2C/D show increased R-loop accumulation in DDX5-depleted cells in the nucleus and at specific genomic loci. Are the primers used in the DRIP-qPCR experiments located at the promoter, gene body or transcription termination regions of these genes? The authors should test different primer pairs along the genes to analyze whether the difference of R-loop levels is more or less pronounced at specific functional elements of these genes (e.g. promoter, gene body and termination regions, see comment above).

REPLY: We indicated the location of R-loop and primers for each locus in the DRIP-qPCR experiments (Figure 1D and Figure 7). These analyses have assisted us in defining the role of DDX5 at transcription termination sites.

• A substantial improvement of the manuscript would be to perform DRIP-Seq in siCTRL, siDDX5 and siPRMT5-depleted cells and map the location and changes of R-loop distribution genome-wide. This would give a much more comprehensive understanding about the role of DDX5 and PRMT5 in R-loop resolution at transcription termination sites. This conclusion is currently based only on a single gene, namely the actively transcribed beta-actin gene. In addition, ChIP-Seq of RNAPII under the same conditions would complement these data and provide a genome-wide view of transcriptional read-through upon DDX5 or PRMT5 depletion. As DRIP-qPCR and ChIP-qPCR appear to be well established in the lab, making libraries of these samples and sequence them should not be out of the scope for this study.

REPLY: We concur that genome-wide analysis would be the best way to go. We attempted DRIP-seq analysis with siDDX5, siPRMT5 and XRN2. Peak calling was a major challenge. Our bioinformaticians, who are trained with ChIP seq sharp peaks, didn't know at first how to analyze the large restriction enzyme digested fragments. Furthermore, for some technical reason our IgG IPs had lots of background and it was difficult for peak calling. All in all, we wasted over 6 months with little new data and thus we opted for a candidate approach with half dozen genes. In sum, we are disappointed in the lack of genome-wide data, but we provide a detailed mechanism demonstrating a functional interaction between DDX5 and XRN2 at transcriptional termination sites regulated by the arginine methylation of the RGG/RG motif of DDX5 in PRMT5 manner.

• Figure 5F shows that XRN2 degrades the RNA only when released by DDX5 in vitro. This is not surprising as XRN2 requires a free 5' RNA-end to allow 5' to 3' exonuclease activity which is not provided in the used R-loop substrate. The authors should address this issue by using a different RNA primer with an extended non-complementary 5' overhang that is not annealed to the DNA. This could give further mechanistic insights whether XRN2 alone can degrade the R-loop substrate even in the absence of DDX5.

REPLY: XRN2 requires a 5'-P end to allow the exonuclease activity to proceed in the 5-3' direction (Stevens & Poole, 1995). Thus, in our R-loop substrate, the 5'-p-end is not protuberant and might not be accessible for RNA degradation. We designed new substrates adding 13 bases non-complementary 5' overhang sticking out of the R-loop. An excess of purified XRN2 (100-400 nM) on the 5' overhang R-loop substrate (100 nM) lead to a progressive degradation of the RNA, leading to faster migrating forms of the substrate (Figure S7C, left). At 400 nM XRN2, the resulting product had a higher mobility than a R-loop with complementary RNA (our original substrate, lane 5). This infers that XRN2 cannot degrade completely the RNA and is stopped at the RNA/DNA junction. To recapitulate this result, we used a simpler RNA/DNA duplex with the same protuberating 5'-p-end (Figure S7C, right). Similarly, XRN2 could not degrade completely the RNA, leading to a higher migrating form than the annealed RNA/DNA duplex without the 5' overhang (lane 5). Altogether these results support our previous experiments where DDX5 needs to unwind the R-loop to allow complete degradation of the RNA by XRN2.

MINOR:

• The BrdU foci experiment shown in Figure S1D should be more thoroughly described in the Materials and Methods section. I assume it's a native BrdU assay after pulse labelling with BrdU to visualize increased amounts of single-stranded DNA but there are no details about the BrdU labelling procedure provided.

REPLY: The BrdU foci experiment in Figure S1D is now described in Materials and Methods.

• One open question is whether arginine methylation of DDX5 mediated by PRMT5 directly enhances the enzymatic R-loop resolution activity of DDX5. The authors should repeat the R-loop

resolution assays and compare the activities of *in vitro* PRMT5-methylated DDX5 (Figure 4C/D) and unmethylated DDX5 on the *in vitro* R-loop substrates (Figure 2B).

REPLY: We have repeated the *in vitro* R-loop resolution assay with the pure recombinant WT DDX5 and DDX5-RK. Both proteins have R-loop unwinding activity i.e. they both unwound R-loops *in vitro* to the same extent (Figure S5). This finding is not unexpected, as both enzymes have an intact DEAD box helicase domain. *In vivo*, the WT DDX5 and DDX5-RK do behave differently as analyzed by DRIP-qPCR (Figure S5H) consistent with our model that the RGG/RG motif of DDX5 recruits XRN2 at transcriptional termination sites for R-loop resolution.

- Are DDX5 and PRMT5's R-loop suppression/resolution activities (Figure 3E/F) related/work together in the same pathway? To address this, the authors should determine R-loop accumulation by S9.6 staining and DRIP-qPCR after double-knockdown of DDX5 and PRMT5.

REPLY: We performed double-knockdown of DDX5 and PRMT5 and DRIP-qPCR analysis (Figure 4B, 4C). To further show that DDX5 and PRMT5 are linked in the same pathway, we performed double knockdown (Figure 4B) and assessed R-loops at 4 loci (*EGR1*, *MALAT1*, *HIST1H2BG*, and *RPPH1* loci). The double depletion did not have a further increase in R-loop than the single depletion of either PRMT5 or DDX5 (Figure 4C), suggesting PRMT5 and DDX5 are functionally linked for R-loop resolution at these loci.

- The efficiency of the knockdown conditions of XRN2 (Figure 5I) and PRMT5 (Figure 5J) are not shown and should be evaluated by Western blot and/or RT-qPCR analysis.

REPLY: The successful knockdown of XRN2 and DDX5 are now shown (now Figure 8C).

- In Figure 5, the authors should also assess the level of R-loops by DRIP-qPCR at the indicated primer locations of the beta-Actin gene in siCTRL, DDX5 and PRMT5 depleted cells to further investigate the interdependence of these factors and their RNA:DNA hybrid resolution activities.

REPLY: We performed DRIP-qPCR at the different locations of the b-actin gene in siCTRL, siDDX5, siXRN2 and siPRMT5 depleted cells. Knockdown of DDX5, XRN2 or PRMT5 led to significant increase of R-loop accumulation at poly (A) site and its downstream transcription termination zone region D (Figure 8C, 8E, poly (A) and RD). Knockdown of DDX5 or PRMT5 but not XRN2 also caused increase of R-loop in the gene body (RB, Figure 8C, 8E), suggesting that DDX5 and PRMT5 both have XRN2-dependent at the transcription termination sites as well as XRN-2 independent functions for R-loop resolution in other gene regions, for example in the gene body.

2nd Editorial Decision

25th April 2019

Thank you for submitting your revised manuscript for our consideration. It has now been seen once more by the original reviewers 1 and 3, whose comments are copied below. As you will see, referee 3 now supports publication, pending a few remaining minor modifications, despite the inability to obtain genome-wide DRIP-seq data. However, referee 1 still retains some more substantive concerns, not only regarding the missing genome-wide picture but in particular also with some of the newly added data.

While I would in light of referee 3's assessment be happy to not insist on inclusion of DRIP-seq data, I do feel that the other issues raised by referee 1 would still need to be satisfactorily addressed before the study should become acceptable for publication in The EMBO Journal. I would therefore be willing to grant an exceptional second round of experimental revision in this case, to allow you to deal with these remaining issues and to strengthen these points through additional, more definitive data.

REFeree REPORTS

Referee #1:

The authors have tried to address my previous comments, though they failed to do the genome-wide DRIP-seq analysis as requested (see comment 1 below). While doing these revisions, the authors have introduced a number of new panels where the quality of the data was not sufficient to draw such conclusions, in particular Figure 8D Pol II Chip and 8F DDX5 ChiP (discussed in comment 2 below). I feel strongly that the authors have to improve the quality of their data (point 2 below) and be precise with the interpretation of their data (point 3 below) prior to consideration of publishing this paper in EMBO J.

1. In relation to previous comment 1: One of my and reviewers 3 requests were to extend the paper by using the genome-wide data, DRIP-seq. The authors state (in the response to the Reviewers 3) that this technique did not work in their hands, even though it is quite a commonly used methodology, successfully employed by many labs. So for me it is difficult to understand why it does not work in the hands of the authors. Also their approaches of using restriction enzymes generating big fragments in DRIP experiments, and not sonication, may not be ideal for the purposes of this study. Instead of genome-wide analysis the authors have examined 6 different genomic loci (new Figure 7). The limitation of this approach is that it does not tell the readers if these effects are genome-wide or if they are characteristic to any particular gene category.

2. In relation to previous comment 2. The Flag DDX5 Chip in Figure 8F is of a very poor quality and massive error bars. Additional experiments should be performed to improve the quality of this experiment. The interpretation of the data in Figure 8D (Pol II Chip with over-expressed Flag-DDX5 WT and RK mutant) is impossible since over-expression of empty pCDNA plasmid in DDX5 KD cells does not recapitulate the phenotype observed in DDX5 kd cells (I.e. transcriptional decrease over the prom; Figure 8C middle panel). Therefore, any conclusions drawn by the authors from this experiment are not justified by the data. The quality of this figure should be improved prior to any conclusions drawn.

3. In relation to previous comment 5. The authors have performed rescue experiment with DDX5 and DDX5-RK mutant followed by DRIP analysis (Figure 5H), however their interpretation of the results are rather biased. As seen from these data, the DDX5-RK mutant did not rescue the DDX5 KD phenotype in EGR1 and RPPH1, however the partial rescue was observed in MALAT1 and HIST1 genes. The authors should clearly state it in the paper and discuss it in the discussion. Their interpretation that a full rescue for all four genes was observed is unsubstantiated by their data.

Referee #3:

This revised manuscript is significantly improved over the original. The authors have mainly addressed the reviewer's comments and have also added significant new data which strengthen the paper's conclusions that a pathway that includes DDX5, PRMT5 and XRN2 has a role for R-loop resolution specifically at transcription termination sites. Even though my major critique has not been addressed to obtain a genome-wide view of regions affected by this pathway, this seems to be due to technical reasons and the authors made a great effort to address my questions at several other independent loci (Figure 7). The results are consistent with the previous data on the beta-actin gene so I think it is unreasonable to ask for more experiments at this point to allow timely publication of the manuscript in the EMBO journal.

Nevertheless, I have two minor concerns that should be addressed prior to publication:

MINOR:

- Figure 1D: The panels showing the R-loop accumulation across the 4 gene regions appear "fuzzy" compared to the panels shown i.e. in Figure 7. The authors should replace them with higher quality images.
- Figure 3B: Can the authors provide an additional merged panel with DAPI staining? Otherwise, it is hard to assess the cellular localization of the measured signals. It appears also surprising that the (presumably) nuclear S9.6 staining is only located at the TA-KR/TetR-KR loci without the typical strong S9.6 nucleolar staining as observed i.e. in Figure 2F. The authors should explain this discrepancy.

Editor/referee response

Referee #1:

The authors have tried to address my previous comments, though they failed to do the genome-wide DRIP-seq analysis as requested (see comment 1 below). While doing these revisions, the authors have introduced a number of new panels where the quality of the data was not sufficient to draw such conclusions, in particular Figure 8D Pol II Chip and 8F DDX5 ChiP (discussed in comment 2 below). I feel strongly that the authors have to improve the quality of their data (point 2 below) and be precise with the interpretation of their data (point 3 below) prior to consideration of publishing this paper in EMBO J.

1. In relation to previous comment 1: One of my and reviewers 3 requests were to extend the paper by using the genome-wide data, DRIP-seq. The authors state (in the response to the Reviewers 3) that this technique did not work in their hands, even though it is quite a commonly used methodology, successfully employed by many labs. So for me it is difficult to understand why it does not work in the hands of the authors. Also their approaches of using restriction enzymes generating big fragments in DRIP experiments, and not sonication, may not be ideal for the purposes of this study. Instead of genome-wide analysis the authors have examined 6 different genomic loci (new Figure 7). The limitation of this approach is that it does not tell the readers if these effects are genome-wide or if they are characteristic to any particular gene category.

2. In relation to previous comment 2. The Flag DDX5 Chip in Figure 8F is of a very poor quality and massive error bars. Additional experiments should be performed to improve the quality of this experiment. The interpretation of the data in Figure 8D (Pol II Chip with over-expressed Flag-DDX5 WT and RK mutant) is impossible since over-expression of empty pCDNA plasmid in DDX5 KD cells does not recapitulate the phenotype observed in DDX5 kd cells (I.e. transcriptional decrease over the prom; Figure 8C middle panel). Therefore, any conclusions drawn by the authors from this experiment are not justified by the data. The quality of this figure should be improved prior to any conclusions drawn.

REPLY: The data of Figure 8C and 8F were repeated and the quality is now improved.

3. In relation to previous comment 5. The authors have performed rescue experiment with DDX5 and DDX5-RK mutant followed by DRIP analysis (Figure 5H), however their interpretation of the results are rather biased. As seen from these data, the DDX5-RK mutant did not rescue the DDX5 KD phenotype in EGR1 and RPPH1, however the partial rescue was observed in MALAT1 and HIST1 genes. The authors should clearly state it in the paper and discuss it in the discussion. Their interpretation that a full rescue for all four genes was observed is unsubstantiated by their data.

REPLY: The following text was added to the results section "*By contrast, the Flag-DDX5-RK mutant fully reversed the effects of siDDX5 at EGR1 and RPPH1 loci, and partially reversed the siDDX5 effects at MALAT1 and HIST1H2BG loci (Figure 5H).*"

Referee #3:

This revised manuscript is significantly improved over the original. The authors have mainly addressed the reviewer's comments and have also added significant new data which strengthen the paper's conclusions that a pathway that includes DDX5, PRMT5 and XRN2 has a role for R-loop resolution specifically at transcription termination sites. Even though my major critique has not been addressed to obtain a genome-wide view of regions affected by this pathway, this seems to be due to technical reasons and the authors made a great effort to address my questions at several other independent loci (Figure 7). The results are consistent with the previous data on the beta-actin gene so I think it is unreasonable to ask for more experiments at this point to allow timely publication of the manuscript in the EMBO journal.

Nevertheless, I have two minor concerns that should be addressed prior to publication:

MINOR:

- Figure 1D: The panels showing the R-loop accumulation across the 4 gene regions appear "fuzzy"

compared to the panels shown i.e. in Figure 7. The authors should replace them with higher quality images.

REPLY: The images were indeed 'fuzzy' and we have replaced them.

• Figure 3B: Can the authors provide an additional merged panel with DAPI staining? Otherwise, it is hard to assess the cellular localization of the measured signals. It appears also surprising that the (presumably) nuclear S9.6 staining is only located at the TA-KR/TetR-KR loci without the typical strong S9.6 nucleolar staining as observed i.e. in Figure 2F. The authors should explain this discrepancy.

REPLY: We now include DAPI staining images to better visualize the nuclei.

We have added the following text to address the S9.6 without the strong nucleolar staining in the results section. *"The signal we observe is similar to published studies (Teng et al., 2018). Under the conditions used (the staining involves a steaming step on a 95 °C heating block for 20 min to expose the antigen and blocking with 5% BSA) the S9.6 focus is more apparent than the typical nucleolar staining observed with standard S9.6 staining protocols. The fact that the cellular system is using a defined locus where multiple breaks are induced by Killer Red, this blocks transcription very efficiently leading to R-loop accumulation and a stronger signal over typical nucleolar staining."*

3rd Editorial Decision

28th May 2019

Thank you for submitting your final revised manuscript for our consideration. I am pleased to inform you that we have now accepted it for publication in The EMBO Journal.

Corresponding Author Name: Stephane Richard and Jean-Yves Masson

Manuscript Number: 2018-100986R